# The Promise of Nanotechnology in Personalized Medicine

**DOI:** 10.3390/jpm12050673

**Published:** 2022-04-22

**Authors:** Maha Ali Alghamdi, Antonino N. Fallica, Nicola Virzì, Prashant Kesharwani, Valeria Pittalà, Khaled Greish

**Affiliations:** 1Department of Biotechnology, College of science, Taif University, Taif 21974, Saudi Arabia; mahaasg@agu.edu.bh; 2Department of Molecular Medicine, Princess Al-Jawhara Centre for Molecular Medicine, College of Medicine and Medical Sciences, Arabian Gulf University, Manama 329, Bahrain; 3Department of Drug and Health Sciences, University of Catania, 95125 Catania, Italy; antonio.fallica93@gmail.com (A.N.F.); nicola.virzi@yahoo.com (N.V.); vpittala@unict.it (V.P.); 4Department of Pharmaceutics, School of Pharmaceutical Education and Research, Jamia Hamdard, New Delhi 110062, India; prashantdops@gmail.com

**Keywords:** nanomedicine, personalized medicine, pharmacogenetics, pharmacokinetics

## Abstract

Both personalized medicine and nanomedicine are new to medical practice. Nanomedicine is an application of the advances of nanotechnology in medicine and is being integrated into diagnostic and therapeutic tools to manage an array of medical conditions. On the other hand, personalized medicine, which is also referred to as precision medicine, is a novel concept that aims to individualize/customize therapeutic management based on the personal attributes of the patient to overcome blanket treatment that is only efficient in a subset of patients, leaving others with either ineffective treatment or treatment that results in significant toxicity. Novel nanomedicines have been employed in the treatment of several diseases, which can be adapted to each patient-specific case according to their genetic profiles. In this review, we discuss both areas and the intersection between the two emerging scientific domains. The review focuses on the current situation in personalized medicine, the advantages that can be offered by nanomedicine to personalized medicine, and the application of nanoconstructs in the diagnosis of genetic variability that can identify the right drug for the right patient. Finally, we touch upon the challenges in both fields towards the translation of nano-personalized medicine.

## 1. Introduction

Personalized medicine may be defined as the tailored individualized management approach to achieve the right drug at the right dose to the right patient [1]. The approach was driven by multiple factors including unjustifiable drug adverse effects in many patients as well as lack of unity in drug efficacy that can vary from 25 to 80% according to drug classes.

Personalized medicine involves proteomic, genomics, and epigenetic studies, as well as specific patient health conditions and environmental influence [2]. In turn, nanotechnology is a broad term that encompasses systems in the range of 10–100 nm [3]. The term also implies the ability to control structures at this nano-range towards a desirable outcome. Molecules at the nano size range could interact with cells at the subcellular and molecular levels as the size allows for this otherwise unattainable interaction at a larger scale (e.g., larger than 1 μm scale). Nanomedicine had been implicated in the prevention, monitoring, diagnosis, and treatment of disease, and many of these inventions are used every day in the current clinical practice [4].

The intersection between nano and personalized medicine lies at multiple points. Firstly, the diagnostic area, and here nanotechnology has a lot to offer in areas of exploring the status of specific drug targets, the pharmacogenetic testing, and the ability to perform both in vitro and in vivo testing. Secondly, the therapeutic area, as the nanomedicine can tailor the drug to a specific target identified for a specific disease in a specific patient [5].

In addition, with nanomedicine, due to its targeting capability, it is possible to achieve much higher doses than the maximum tolerated dose for the non-formulated drug. Hence, the dose can be tailored based on individualized patient conditions [6]. Finally, nanomedicine can circumvent two major determinants in individualized drug response related to the variability in cytochrome-P enzymes (CYP) and drug transporters in different populations (Figure 1). Nanomedicine drug formulation could effectively render formulated drugs as stealthy to metabolizing enzymes as well as make it intracellular in the endocytic process, which is independent of the transporter.

In the following sections, we overview the current personalized therapeutics in clinical practice then discuss the advantages that can be offered by nanomedicine for the design of personalized medicine, and then we discuss the potential use of various nanoparticles in pharmacogenetic approaches to predict patients’ response to treatment.

## 2. Current Personalized Therapeutics in Clinical Practice

The United States Food and Drug Administration (FDA) labeled 486 drugs that require specific pharmacogenetic testing. Each drug has its own biomarker to predict the response according to genetic profile, and in many instances indicates the use of the drug and/or the drug dose relevant to biomarker results [7]. Personalized medicine covers many areas of clinical disciplines, such as oncology, neurology, psychiatry, anesthesiology, hematology, cardiology, and gastroenterology (Figure 2a).

The biomarkers’ majority is related to CYP enzyme polymorphisms, resulting in patients having different metabolic activities as described later. Most of the CYP variations are found in CYP2D6; CYP2C9 related to the therapeutic area of anesthesiology; CYP2D6, CYP2C19, CYP2C9, and CYP3A5 related to the therapeutic area of cardiology; CYP2C9 in hematology; and CYP2D6 in oncology (Figure 2b–d,f).

The second phase metabolic enzymes are also used as biomarkers such as transferases that comprise uridine disphosphate glucoronosyltransferase (UGTs). UGTs are a major part of phase II metabolism and are endoplasmic reticulum-bound enzymes responsible for the process of glucuronidation that includes 22 different functional enzymes.

In addition, glutathione S-transferases (GSTs) and sulfotransferases (SULTs) are important conjugative enzymes mediating phase II reactions. In the list of biomarkers, N-acetyltransferases 1 and 2 (NAT1 and NAT2) cytosolic enzymes catalyze the acetylation reactions; thiopurine S-methyltransferase (TPMT) is a cytoplasmic enzyme that catalyzes the S-methylation of drugs. Efflux transporters include ATP-binding cassette (ABC) and solute-linked carrier (SLC) proteins. These biomarkers were found to be related to some therapeutics in cardiology, neurology, and oncology (Figure 2c,e,f) [8].

The majority of drugs associated with biomarkers are anticancer drugs (206 drugs) (Figure 2a) and mount to 55% of total personalized drugs.

For instance, there are 20 anticancer drugs that target ERBB2(HER2). The situation of HER2 biomarker would influence the prescription, dosage, or safety of administered drugs such as Abemaciclib, Lapatinib, Alpelisib, Neratinib, Trastuzumab, and Olaparib.

## 3. The Pharmacokinetics (PK) and Pharmacodynamics (PD) Properties of Nanomedicine and Formulation Advantages

Nanodrugs were originally designed to improve the properties of an already available drug or diagnostic agent. Today nanoparticles are designed to minimize local and systemic side effects, to enhance the bioavailability of drugs taken orally, and to improve the half-life and overall pharmacokinetic and pharmacodynamic properties. Moreover, nanoparticles can reduce the frequency of administration of drugs, leading to better compliance and ameliorating clinical outcomes [9,10].

Compared with traditional drug delivery systems, there are several pharmacokinetic advantages that a nanodrug can offer: major solubility and absorption, the possibility of controlled release, improvement of drug stability and metabolism, reduction of side effects, extended blood circulation, and better performance in targeted delivery [11].

Although the focus of pharmaceutical industries and of nanoparticles development remains the optimization of the one-size-fits-all solution, the ability of nanomedicine to personalize the pharmacokinetics and pharmacodynamics of the drugs and to overcome the biological barriers limitations represents for an individual or a cohort with specific genome requirements a promising personalized therapeutical opportunity [12,13]. Thus, the aim of nanoparticle usage in personalized medicine is to exploit specific genomic patient information, comorbidities, and subjection to the environment to create an individualized treatment with improved drug specificity and optimized doses delivered in a specific site [13]. Regarding the pharmacokinetic processes, a drug could undergo four stages, indicated by the acronym ADME: absorption, distribution, metabolism, and excretion (Figure 3). The nanoparticle advantages in pharmacokinetic and pharmacodynamic processes will be discussed in the following sections.

### 3.1. Absorption

The absorption process starts with the entering of nanodrugs into blood circulation via different physiological routes. The frequently used routes of administration are intravenous, oral, transdermal, and nasal administration. The most important and extensively used routes are the intravenous and the oral route. By means of an intravenous injection, there is no need for absorption processes, indeed the nanodrug directly enters the blood circulation. The surface charge, hydrophobicity, and the size of nanoparticles affect their mucosal absorption. Indeed, smaller nanoparticles could have a better transcellular uptake when compared to larger ones [14]. Moreover, nanoparticles with a size up to 500 nm generally have better circulating and targeting ability and display a safer profile, reducing the risk of capillary occlusions and embolism [15]. In addition, larger nanodrugs are rapidly cleared from the bloodstream due to opsonization processes and to the action of macrophages of the reticuloendothelial system (RES) [9]. Via non-intravenous administration routes, nanodrugs must pass through biological barriers before reaching the bloodstream. This process is not always possible for traditional drugs, due to their unsuitable chemical properties such as worst log P, solubility, and stability. Nanoparticles can overcome these issues, improving the passage through biological barriers and absorption, allowing the use of different administration routes that could not be used for traditional drugs [16].

For instance, after oral administration, a traditional drug could be degraded by the low gastric pH or enzyme activity or could not be soluble in human fluids, affecting the absorption process and the therapeutic action. When a drug is encapsulated in nanoparticles, all these issues are overcome, and the absorption process strictly depends on the nanoparticle’s physicochemical properties. When nanoparticles are orally administered, they could be absorbed through the gastrointestinal tract by different processes, such as paracellular pathway transport, transcytosis mediated by the carrier, passive cross-cell diffusion, or microfold cells (M cells) absorption [16,17]. Thus, nanoparticles are able to enter the systemic blood circulation by intestinal lymph node or through the portal vein; moreover, nanoparticles absorption by M cells improves the drug bioavailability because of the bypass of cytochrome P450 metabolism, hepatic first-pass, and P-glycoprotein (P-gp)-mediated efflux [18].

One example of the advantages of nanoparticles in overcoming poor absorption is the anticancer drug Olaparib (Ola). The drug’s main action is through poly ADP ribose polymerase (PARP) inhibition. The drug showed selective activity in tumors with a mutated BRCA gene. However, Ola is poorly absorbed through the gastrointestinal tract. Ola entrapment into a liposphere efficiently improved its oral bioavailability and further reduced its hematological toxicity [19].

The skin is the most difficult obstacle to overtake for nanodrugs designed for a transdermal administration. The skin is composed of different lipophilic and hydrophilic layers, and this variability influences the difficult traditional drug absorption [11]. However, Wang et al. proved that imidazole-based ionic liquid microemulsions are able to reduce the skin barrier properties, disrupting the arrangements of corneocytes and moderating the surface characteristics of the stratum corneum [20].

During the nasal administration, nanodrugs deposited in the lungs can be removed via mucociliary clearance to the gastrointestinal tract, exhaled or sequestered, and degraded by macrophages. The remainder can cross the mucus and lung epithelium, being absorbed [21]. In addition, it is known that positively charged nanodrugs have an enhanced absorption process through the lung mucosa [17], because of the interaction with the negatively charged sulphate sialic acid and sugar moieties of mucin [14].

### 3.2. Distribution

The distribution process starts when the nanodrug is translocated from the blood circulation to the tissues and cells. Usually, after absorption, nanoparticles are rapidly distributed and accumulated to the spleen, bone marrow, and liver, because of the presence of the sinusoidal endothelial capillaries. Instead, the nanoparticles amount is low in the kidney after injection and highest after 1 month of the administration [22]. Moreover, the distribution of nanodrugs to the brain tissue is difficult to achieve because of the blood–brain barrier (BBB). Nanoparticles usually have a size greater than 5 nm, and this influences the distribution because the majority of the endothelia present in the human body have fenestration of about 5 nm [17]. It is possible to distinguish between two different types of distribution processes for nanodrugs, which are known as passive and active targeting.

#### 3.2.1. Passive Targeting and EPR Effect

The passive targeting essentially depends on the size of nanoparticles and on the intact fenestration of the endothelia. The enhanced permeability and retention effect (EPR) is a phenomenon discovered by Maeda et al. [23] that allows a more specific drug accumulation in solid tumors and in infection sites [24].

On the one hand, tumor blood vessels and inflamed tissues have an impaired lymphatic drainage system and leaky vasculature systems with pore sizes that vary from 200 nm to about 800 nm, depending on the cell type or the condition of the tissues. This is mainly due to a defective vascular architecture and widespread and rapid angiogenesis. On the other hand, normal tissues have a vasculature system with pore sizes that do not allow the nanoparticles to pass. For this reason, nanodrugs that fulfill the dimensions reported above can passively arrive at the impaired site and release the drug, having a more specific and effective therapeutic action, with a massive reduction of the needed dosage and consequently reduced onset side effects [25].

Greish et al. utilized styrene-co-maleic acid (SMA) as a micellar nano-carrier, for many anticancer agents. In a study utilizing SMA-doxorubicin, the group demonstrated the preferential accumulation of SMA-doxorubicin 13-fold higher in tumor tissues compared to equivalent doses of free doxorubicin. Similarly, the group utilized the same micellar system to deliver nano micelles containing dasatinib and targeting PDGF, KIT, and ABl [26,27].

At the same time, it is possible to say that the EPR effect could have some limitations. Indeed, non-solid tumors such as leukemia cannot benefit from the EPR effect. Moreover, different tumor types could have different pore sizes, and the nanodrug may be able to target only some areas of the tumor, giving an unpredictable and maybe inefficient therapeutic outcome [28].

#### 3.2.2. Active Targeting

Although EPR effect helps nanoparticles selectively reach the tumor interstitium, it has no efficiency in promoting cellular uptake. The functionalization of nanoparticles with ligands able to address the nanodrugs to specific cancer cells or sub-cellular sites not only reduces the side effects due to passive targeting but also promotes the uptake of the nanoparticles into the cells (Figure 4) [29]. This is due to the presence/overexpression of specific receptors over the cell surface that can be targeted by means of a specific binding moiety. Moreover, active targeting could be a suitable method to exploit in personalized medicine, allowing to treat the patient using the best possible ligand to selectively direct the right dose of nanodrug to the right site.

A wide range of ligands was used to decorate the surface of nanoparticles, including antibodies or their fragment, aptamers, peptides or proteins, and many other receptor ligands.

##### Ligand-Based Targeting

A large variety of ligands were used to decorate the nanodrug’s surfaces, such as adenosine, folate ligands, and glucose.

Adenosine is a nucleoside that plays a fundamental role in cellular function regulation and activation of the adenosine receptors. Various types of tumors, such as colorectal, prostatic, lymphoma, and breast cancer overexpress the adenosine A1 receptor [30]. Swami et al. investigated the ability of adenosine to direct solid lipid nanoparticles (SLN) charged with docetaxel into human breast cancer and prostate cancer [31]. The adenosine-conjugated SLN charged with docetaxel (ADN-SLN-DTX) demonstrated higher cytotoxicity and better pharmacokinetic parameters compared to the unconjugated SLN-DTX.

Folate receptors belong to a high-affinity folate-binding protein class that plays a role in the cellular uptake of folate. While normal tissues do not present any folate receptors on surface cells, they are overexpressed in several types of solid tumors such as testicular, brain, endometrium, breast, and colon, and this overexpression is usually related to poor clinical outcomes [30,32,33]. As reported by Patil et al., the grafting of pegylated liposomes charged with a mitomycin C prodrug with folate ligands leads to a higher level of nanodrug uptake by tumor cells, leading to increased cytotoxicity [34].

Glucose could be exploited for the active targeting approach as well. Cancer cells are metabolically more active compared to normal cells, and this leads to an increased need for glucose. For this reason, glucose-coated nanodrugs, thanks to the presence of the glucose transporter channel 1, have a higher permeation into cancer cells and this could be helpful for the theragnostic approach [35]. In this context, Gromnicova et al. reported that glucose-coated gold nanoparticles are able to transport loperamide selectively and efficiently across the BBB [36].

The use of a ligand-based targeting approach has many advantages since ligands compared to antibodies or aptamer ligands are usually much less expensive and easier to conjugate by relatively easy chemical reactions. They are non-immunogenic and safe. Moreover, endogenous ligands are usually released into the cytoplasm or retained into a vesicle. Instead, protein ligands could be directed to a lysosome for their degradation [37].

##### Protein–Based Active Targeting

Several proteins and glycoproteins are able to bind and activate cellular receptors, as in the case of transferrin. Transferrin is a serum glycoprotein that binds transferrin receptors, playing a role in iron transport. In cancer and metastatic cells, transferrin receptors are 100-fold more upregulated, and this makes transferrin an optimal ligand for the active targeting approach [37]. As reported by Cui et al., transferrin-decorated nanodrugs including doxorubicin and curcumin displayed a stronger antitumor effect and decreased the unwanted cytotoxic effect thanks to the achievement of an efficient targeted delivery [38]. Another study reported the ability of human serum albumin nanoparticles functionalized with an antibody directed towards the transferrin receptors that were able to efficiently transport loperamide across the BBB [39].

Antibodies, thanks to their fragment antigen binding (Fab), possess high specificity and affinity toward surface cell receptors. Among these, it is dutiful to list the epidermal growth factor receptor 2 (HER2) targeted by trastuzumab. As reported by Arya et al., the conjugation of trastuzumab with gemcitabine-loaded chitosan nanoparticles can have a superior antiproliferative and cytotoxic activity in comparison with the unconjugated ones for the treatment of pancreatic cancer [40].

In addition, the use of the anti-CD20 antibody rituximab as an active target ligand improved the therapeutic efficiency of a nanodrug directed toward chronic lymphocytic leukemia cells. For instance, poly(lactide–co–glycolide) (PLGA) nanoparticles conjugated with rituximab and loaded with nutlin-3 were able to selectively target JVM–2 B leukemic cells, determining antiproliferative effects through the activation of p53 [41].

Antibody fragments such as fragment antigen-binding (Fab) and single-chain variable fragments (scFV) could be used instead of the entire antibodies. The smaller dimensions of the fragments could allow multiple attachments of antibody fragments, improving the specificity and therapeutic effect.

For instance, Sapra et al. proved that stealth immunoliposome (SIL) formulations of doxorubicin and vincristine conjugated with anti-CD19 Fab fragments had longer circulation time and better therapeutic outcomes than the ones conjugated with the entire antibody [42]. Moreover, the absence of the crystallizable fragment region (Fc) decreases the immunogenicity and the uptake by RES, ameliorating the pharmacokinetic profile [37].

##### Aptamers, Gapmers and siRNA Active Targeting

Aptamers are single-stranded RNA or DNA oligonucleotides able to specifically bind proteins or biological targets and are synthesized by means of an in vitro process called SELEX (systematic evolution of ligands by exponential enrichment). Their high sensitivity and selectivity for their target make them a valid alternative to antibodies for the active targeting approach, with some advantages. As an example, aptamers are less immunogenic and much easier and cheaper to produce. Moreover, they allow the use of routine synthetic chemistry reactions for the bioconjugation and cause better penetration in tissues, with consequent improved therapeutic effects.

Lopez-Nunes et al. proposed the utilization of gold nanoparticles functionalized with an AS1411 G-quadruplex DNA aptamer for the targeted drug delivery of the proapoptotic molecule imiquimod for the treatment of cervical cancer [43]. The chosen aptamer possesses a strong affinity for nucleolin, thus it is internalized by tumor cells and releases the payload selectively in the diseased tissue. Results showed that this aptamer-based strategy was successful in reducing HeLa cell viability and had less side effects on healthy cells thanks to their lower expression of nucleolin and consequent reduced interaction between this protein and the AS1411 DNA aptamer.

A similar strategy was exploited also for the management of lung cancer. Indeed, Zhang and co-workers reported the development of a stimuli-responsive EGFR aptamer-modified PLGA-SS-PEG nanoconstruct loaded with homoharringtonine, a natural alkaloid with antitumoral properties [44]. The interaction of the aptamer with the EGFR protein up-regulated in non-small lung cancer cells (NSCLC) allowed the internalization of the nanoparticle by endocytosis. The high reductive intracellular environment promoted by the higher expression of glutathione (GSH) in NSCLC cells triggered the breakage of the disulfide bonds of the nanoparticle, releasing the cargo inside the cell. Results obtained in vitro and in vivo showed that this approach was more effective in decreasing cell viability when compared to the single administration of the cytotoxic drug. Similarly, conjugation of the A10 RNA aptamer, directed towards the prostate-specific membrane antigen (PSMA) with poly (D, L-lactic-co-glycolic acid)-block-poly (ethylene glycol) (PLGA-b-PEG) nanoparticles encapsulating docetaxel showed an improved cytotoxic effect when compared to the unconjugated ones [45].

Gapmers are short chimeric antisense oligonucleotides made of a central DNA-core flanked by 2′-O-methylated-RNA-like sequences. This chimera can bind a complementary oligonucleotide and silence a specific target RNA through its degradation by the action of RNase-H [46]. Gapmers possess an intriguing therapeutic potential in the context of cancer and precision medicine. For example, Garcia-Garrido et al. designed gold nanoparticles coated with citric acid, functionalized with polyethylene glycol (PEG), and branched with short polyethylenimine (bPEI) chains. This scaffold was further cross-linked with succinimidyl 3-(2-pyridyldithio) propionate (SPDP), affording the insertion of a GSH-sensitive disulfide bond in the nanoconstruct structure. Overall, the chemically modified gold nanoparticle showed a high stability and a global positive surface charge, which was exploited for cell transfection and for the formation of electrostatic interactions with gapmers targeting the p53 mutant protein in pancreatic and breast cancer cells [47]. In the presence of GSH, the nanoparticle structure undergoes a degradative process that allows the release of the p53 targeting gapmer. This approach was successful in reducing cell proliferation in PANC-1 and MDA-MB-231 cancer cell lines, which present the p53 mutation, whereas no effects were observed in the MCF-7 cancer cells. Moreover, silencing of the mutated p53 protein reverted the cells susceptibility to gemcitabine, a chemotherapeutic agent whose biological effects are impaired when the p53 mutation is observed. In general, this approach could be particularly useful for patients whose resistance to antitumoral drugs is dependent on the alteration of proteins involved in pro-apoptotic pathways.

The utilization of siRNAs represents another efficient gene-silencing strategy that can be used in antiviral and antiumoral therapeutic contexts. As an example, Idris and collaborators proposed the intravenous use of stealth liposomes as drug delivery systems for siRNAs targeting highly conserved regions of SARS-CoV-2 with the aim of blocking virus expression and replication [48]. These liposomes were formulated by fine tuning the amount of the cationic molecules 1,2-dioleoyl-3-trimethylammonium-propane (DOTAP) and MC3 in order to reduce the toxicity due to an excessive cationic surface charge and in turn ameliorate the endosomal release of the siRNA in lung cells.

A phase 0 clinical study (NCT03020017) highlighted the potential of siRNA utilization for the treatment of recurrent glioblastoma (GBM) [49]. The therapeutic treatment usually involves the utilization of lomustine, carmustine, temozolomide, or bevacizumab, coupled with surgery and radiotherapy. However, this type of cancer is particularly aggressive, and poor chances of survival have often been reported. Therefore, new anticancer strategies based on the discovery of novel small molecules or nanomedicine approaches are urgently needed [50]. Kumthekar et al. developed a RNA interference-based spherical nucleic acids (SNAs), made of a gold nanoparticle conjugated with siRNA oligonucleotides targeting the highly expressed GBM oncogene Bcl2Like12 (Bcl2L12). After an intravenous administration, the SNA construct was able to reach the tumor site as highlighted by the detection of gold through X-ray fluorescence microscopy. Inductively coupled plasma mass spectrometry (ICP-MS) allowed to quantify gold plasma concentration. In general, gold clearance was slower than siRNA clearance, meaning that the nanoparticle slowly releases the cargo at the tumor site. The siRNA was able to efficiently silence the Bcl2L12 oncogene, induce caspase-3 activation, and increase wild-type p53 protein expression. Finally, toxicological studies showed that the nanoconstruct did not determine severe side effects in the treated patients. Despite the fact that additional studies need to be performed, the proposed SNA nanoconjugate denotes the goodness of the siRNA therapeutic efficiency as a promising precision medicine strategy for the treatment of recurrent GBM.

### 3.3. Protection against Degradation Enzymes and Metabolism

The oral administration route suffers from several pharmacokinetic variabilities due to the pre-uptake metabolism and first-pass metabolism [51,52]. Orally administrated drugs have to overcome several host gastrointestinal obstacles such as pH mutability, gastrointestinal motility, physical barriers such as mucus and mucosa, bacterial diversity, and especially several degradation enzymes (e.g., trypsin, lipase, and CYP450) and efflux pump (e.g., P-gp) [53]. Among them, the CYP450 family, and especially CYP3A4, contribute to the large variability in drug response in a special population (e.g., children, pregnant women, elderly, and ethnicities) [54], leading to unpredictable therapeutic results and/or several side effects. The CYP3A4 is responsible for the metabolism of more than 50% of marketed drugs [55] and the amount of this isoenzyme in the gastrointestinal tract is about 40% of those present in the liver [52]. Therefore, the pre-uptake metabolism of orally taken drugs should not be overlooked. CYP3A4 content in the gastrointestinal tract decreases from the proximal duodenum to the distal ileum, and the activity of the enzyme could largely vary from person to person [56]. Indeed, CYP3A4 content could be influenced by diseases such as obesity, cancer, infection, and inflammation as the CYP3A4 expression strongly depends on the pregnane X receptor (PXR). Indeed, PXR is downregulated in the diseases mentioned above [57].

Many types of nanoparticle formulation strategies have been tested to overcome the pre-uptake metabolism and the variability derived from it, in order to enhance the pharmacokinetic profile of the oral administrated drugs. The use of mucoadhesive polymers could allow a better residence time and resistance to peristaltic movement, favoring the absorption of the nanodrug through the gastrointestinal tract. For instance, Han et al. enhanced the adsorption at the mucus layer by coating the alendronate containing liposomes with the cationic polysaccharide chitosan, which is able to interact with the negatively charged mucin present in the proximal tract [58].

Another strategy is the use of highly lipophilic lipid nanoparticles (HLLN), especially those containing triglycerides, which are able to transport the drug across the enterocytes and to lymphatic vessels. Indeed, the HLLN is degraded in the intestinal lumen, absorbed by enterocytes, and successively recomposed into chylomicron. Then, the drug incorporated into the chylomicron will be sorted directly into the lymphatic vessels avoiding pre-uptake metabolism and first-pass metabolism [59].

The use of nanoparticles containing CYP3A4 inhibitors to inhibit the isoenzyme during transcytosis could be another option [60], but the risk of potential toxic side effects due to shut-down of the physiological action of the enzyme could be a relevant downside [61]. On the contrary, the use of common surfactant, co-solvents, and oil such as Tween, PEG, Poloxamer, Cremophor, Polysorbate 80, and oleic acid proved to be able to notably inhibit the CYP3A4 activity [62,63].

Moreover, another delivery approach is the use of M cell-targeting nanoparticles to directly deliver the nanodrug into the lymphatic vessels. The M cells are located in the Peyer’s patches and have a high inclination to transport and induce endocytosis of antigens into these ones. Strategies that target the M cells include the mimicking of the entry of pathogens such as Salmonella and Yersinia or the targeting of specific receptors such as the integrins that are located on the surface of these cells [64].

Another strategy that may be used is the development of nanoparticles capable of releasing the drug in low CYP3A4 expression areas in pH-dependent manner (e.g., in the ileum). The distal jejunum and ileum are regions in which the CYP3A4 expression and activity are lower than that present in the proximal gastrointestinal tract. As the pH of the jejunum and ileum is near 7 and 8, respectively, the use of pH-sensitive nanoparticles could be useful to deliver a higher concentration of drugs in these regions, in order to over-saturate the CYP3A4 enzyme, and to have a greater number of drugs that are able to bypass the metabolism [64].

Lastly, the use of nanoparticles linked with vitamin B12 receptor–ligand could be exploited in order to avoid the CYP3A4 metabolism. Liu et al. developed polymeric nanoparticles (H/VC-LPNs) integrated with vitamin B12-modified chitosan, which is able to enhance the oral bioavailability of curcumin [65].

As previously stated, the metabolic action of the degradation enzymes of the CYP450 family is one of the most influential factors in the pharmacokinetic destiny of a drug and of its side effects. The gene variability of CYP450 contributes even more to the unpredictability of the administrated drug, usually leading to severe side effects. For instance, this is the case of epilepsy treatment, with it being one of the most studied cases in which the CYP450 genetic factors variability influences the efficiency and safety of the antiepileptic drugs a lot. Thus, the seizure control and the adverse reaction responses are different across the patients, and the large variety of genotypic and phenotypic heterogeneity complicates the physician choice, which, in these cases, relies on empirical data. Most of the metabolism undergone by antiepileptic drugs is mediated by the CYP2C9 [66]. In the isoenzyme, several allelic variants could exist that encode for several isoforms with different metabolic activity, distinguishing between:Poor metabolizers: in which the drug is metabolized very slowly, experiencing several side effects at standard doses;Intermediate metabolizers: in which the drug is metabolized at a slow rate, having potential side effects at standard doses;Extensive metabolizers: in which the drug is metabolized at a normal rate and with minimum risk of side effects and maximum therapeutic efficacy;Ultrarapid metabolizers: in which the drug is rapidly metabolized and removed too quickly to provide a therapeutic effect.

Therefore, the gene testing for the CYP450 enzyme polymorphism could be helpful to identify the level of metabolic activity of the specific phenotype, to classify the patient on the basis of the type of the polymorphism, allowing the use of the right dosage for the seizure control. Established evidence showed how polymorphic CYP2C9 variant allele can notably lead to the various antiepileptic drug concentration levels in the blood [67]. For instance, phenytoin metabolism depends on CYP2C9 activity. Patients with CYP2C19*17 variant allele were found to demonstrate the fast metabolization of phenytoin, resulting in the absence of a drug response [68,69].

In such cases, nanomedicine could be helpful in overcoming these issues, protecting the drug from the degradation enzymes, improving its half-life, and carrying the right dose to the target site.

Nanoformulations of anti-retroviral drugs (ARVs) are a good example of how nanomedicine could improve the bioavailability of a drug, bypassing the CYP metabolism and overcoming the limitations and the side effects of the canonical therapeutic scheme, which includes the need for a pharmaco-enhancer that could lead to drug–drug interactions [70]. For instance, the potent HIV protease inhibitor Atazanavir suffers from a rapid CYP3A4 first-pass hepatic metabolism, leading to low availability. Chattopadhyay et al. demonstrated in vitro how solid lipid nanoparticles encapsulating Atazanavir were able to enhance the uptake of the drug into hCMEC/D3 cell line, also bypassing efflux transporters [71]. Chaowanachan et al. demonstrated how PLGA nanoparticles loaded with Efavirenz (NP-EFV) led to a 50-fold reduction in the 50% IC50 in comparison with the free drug and potent protection against HIV–1 BaL infection in vitro [72].

Considering the information discussed above, it could be possible to state that nanoparticles may have the potential to conduct canonical therapeutic schemes to a more efficient choice for the patient and clinician, moving a step forward towards the future of personalized medicine.

### 3.4. Nanoparticles Interaction with the Microenvironment

Precision medicine’s purpose is to use specific genetic, environmental, and comorbidities in patient information’s in order to perform accurate patient stratification and to treat their disease condition specifically. As previously discussed, nanoparticles could be helpful in precision medicine because of their ability to deliver in a more safely and targeted way the encapsulated drug. Despite these advantages, nanoparticle efficacy is more often diminished because of the heterogeneity of biological barriers and microenvironment of the human body tissues, especially in the case of variability given by the morbidities and comorbidities [13]. Nevertheless, nanoparticle clinical trials are still performed in unstratified patient populations [73] and the tendency will possibly change in the future because of the need for personalized treatment. In oncology, patient stratification proved to be essential to produce positive results, even when patients were treated with nanonmedicine [74]. Hence, patient stratification and nanoparticles modifications based on the patient information have to go hand in hand with successfully performing personalized medicine [75]. Moreover, despite the benefits given by the active targeting, diseased cells markers can vary among patients making target selection processes limiting. Moreover, the microenvironment seems to heavily influence the successfulness of drug-delivery processes [76]. For instance, Qin Dai et al. reported that nanoparticles grafted with antibodies are able to target only 2% of the tumour cells, causing a failure in treatment [77]. Indeed, nanoparticles have to cross the local microenvironment to successfully deliver the drug in the target cells, and here the obstacles may include physical barriers and changes in chemical conditions. Thus, an in-depth understanding of the microenvironments seems to be critical to reach the desired tissues with nanomedicines [13]. Moreover, microenvironment features are generally different from the circulation ones, resulting in the possible alteration in stability and properties of the nanoparticles.

For instance, several components of the tumor microenvironment, such as the extracellular matrix (ECM) density, vasculature, and interstitial fluid, seem to contribute to the non-penetration of nanoparticles in the tumor cells [13,78,79,80]. Moreover, pH or temperature variation in the microenvironment, such as in tumor conditions or in the wound healing process, could negatively influence the destiny of the nanoparticles. On the other hand, these particular conditions may be exploited to perform personalized release of the drug only in the diseased tissues, by means, as example, of pH or temperature-sensitive nanoparticles [78].

Generally, only 0.1% of the free drug is able to accumulate to the target site and about 15% of the administered nanoparticles are able to do the same. This increase in tumor accumulation is usually attributed to the EPR effect, as previously discussed. Many recent findings have reconsidered and greatly de-emphasized the role of EPR effect in the accumulation, proving that only a small fraction of penetrated nanodrugs could be attributed to the EPR effect. Indeed, a crucial role seems to be played by immune cells interaction and protein coronas mechanism [81].

Moreover, the heterogeneous formation of vasculature around the tumor can be strictly influenced by individual factors such as lifestyle, genetics, age, chemotherapy, and comorbidities. To perform personalized treatments, nanoparticles must be selected on the basis of the individual vasculature of the patient [73,82]. In addition to this, Sykes et al. reported that variation in the tumor volume can influence the penetration and accumulation of the nanoparticles in the tumor site. Suggesting that, nanoparticles can be potentially personalized according to the tumor conditions to achieve hopeful therapeutic outcomes [79]. Moreover, in the microenvironment, cells can overproduce altered ECM components, resulting in a denser barrier that obstructs the penetration of nanoparticles [77,80,83]. An additional obstacle for positively charged nanoparticles is the possible charge interaction with the negatively charged ECM components, blocking the permeation into the target site [84,85]. Obviously, the limited nanoparticle perfusion in the brain can be likewise correlated to the limited extracellular space present in the brain microenvironment and to a non-specific adherence to ECM [13].

Additionally, biofilms and mucus layers can influence the distribution of nanoparticles, entrapping them in various mesh pore size or by means of non-specific interaction, leading to clearance from the epithelial surfaces [86]. Mucus composition, viscoelasticity, and hydration depend on physiological conditions and location [86,87,88]. For instance, in cystic fibrosis, the overexpression of MUC5B polymers results in decreasing the mucus clearance and pore sizes [88,89]. Henceforth, since the microenvironment seems to be critical for the destiny of a nanoparticle, it is important to design novel types of nanoparticles or modify them in order to take advantage of this variability. Exploiting endogenous triggers such as the presence of a high level of matrix metalloproteinases (MMps), proteases or of hypoxic, or an acidic microenvironment could enhance nanoparticles degradation and drug release [90,91]. The use of exogenous triggers such as near-infrared light, radiofrequencies, or magnetic fields could also be used to control the nanoparticles delivery from the outside [91,92].

Even the incorporation of macrophage or leukocyte cell membranes derived from the patient into nanoparticles seems to improve the efficiency in the targeted cancer cells, while a weak targeting is given when the donor is different from the patient [93,94]. Usually, nanoparticles wrapped with cell membranes show a massive increase in drug activity compared to a free drug [93].

Another used strategy to overcome the microenvironment-related issues is the use of nanoparticles aimed at the remodeling of the microenvironment. Microenvironment remodeling could be helpful to increase nanoparticle penetration and to sensitize the tumor to a specific treatment. For instance, Wilson et al. proved that the regulation of the gene TREX1 in endothelial cells by means of microRNA can alter the tumor vasculature, sensitizing the tumor to chemotherapy [95]. Moreover, the microenvironment modification could allow to reduce the patient variability and to recruit more eligible patients in the stratification.

In conclusion, it is possible to speculate that precision medicine strictly relies on stratified patient populations, and the improvement of the delivery through the microenvironment could increase the efficacy of the treatments. Targeting the microenvironment could be possible to diminish the differences between patients’ variability, allowing them to be included into stratified populations.

Moreover, the large modification availability for nanoparticles in terms of shape, size, charge, surface properties, and active-targeting ligand modifications may be helpful in precision medicine to better adapt the delivery systems to the microenvironment.

### 3.5. Intracellular Internalization and Subcellular Organelles Targeting

The effectiveness of nanomedicine therapeutic strategies is strictly dependent on the design of the nano formulation since by modulating nanoparticle physicochemical properties it is possible to discriminate between healthy and diseased cells and predict the mechanisms of cellular uptake and intracellular targeting. However, the translation from in vitro to in vivo studies and from theory to practice is often affected by the complexity of the biological systems with which nanoparticles interact [96]. In personalized medicine, these factors need to be carefully taken into consideration, indeed, drug targeting could be highly affected on the basis of age, sex, target tissues and organs, metabolic differences, and concomitant diseases that can alter different biological parameters. For instance, after systemic administration, nanoparticles are often altered in their outer surface properties by the absorption of serum proteins that can mask the presence of ligands grafted in the nanoparticle surface necessary for a specific cellular targeting. The absorbed proteins, known also as protein corona, could drastically change the characteristics of the nano formulation with consequent hampered cellular uptake, inactivation, and premature elimination [97]. The impact of protein corona formation can be partially overcome by the engineering of nanoparticles possessing a zwitterionic charge in their surface and a global hydrophobic nature [98] or by the insertion of poly(ethylene glycol) (PEG) chains with low molecular weight [99].

Bertrand and co-workers tried to understand how the physicochemical properties of PEG-PLGA nanoparticles and their interaction with plasma resident proteins could modulate the biodistribution and clearance [100]. The authors found that a low PEG density reduces the nanoparticle clearance whereas a higher PEGylation determines an opposite effect. Interestingly, a total number of 20 PEG chains per 100 nm^2^ represents a threshold value beyond which the circulation time and clearance of the nanoparticle remain almost the same independently from the nanoparticle size. In addition, this threshold value was found to be consistent in organisms possessing different protein phenotypes. On the other hand, the nature of the proteins composing the protein corona varies on the basis of the steric properties of the nanoparticle surface, with a consequent different impact on the circulation time. Indeed, despite proteins of the complement cascade not seeming to be involved in modifying the clearance of nanoparticles as shown by wild type and complement protein 3 (C3) knockout mice, different results were obtained with apolipoprotein E (ApoE). The latter belongs to a class of proteins hardly adherent to the nanoparticle surface. A low PEG density was associated with a high deposition of ApoE in vivo, with a consequent reduced clearance. An opposite effect was noticed in high PEG-covered nanoparticles or in ApoE^−/−^ animals with a resultant higher nanoparticle opsonization and elimination. The LDL receptor (LDLR) also showed to interfere with nanoparticles distribution and elimination rate independently from the PEG density. In fact, LDLR^−/−^ knockout mice or pretreatment of the experimental animals with a strong LDLR binder such as proprotein convertase subtilisin/kexin type 9 (PCSK9) increased the nanoparticle circulation time when compared to the control.

Recently, the formation of a personalized protein corona started to be considered not only as an obstacle for proper nanoparticles targeting but also as an emerging tool for the development of diagnostic and therapeutic personalized nanomedicine. In this context, Ren and co-workers reported a proteomic study in which polyanionic gadolinium metallofullerenol (Gd@C_82_(OH)_22_) nanoparticles were used for the analysis of the composition of the protein corona in 10 patients with lung squamous cell carcinoma [101]. Interestingly, results led to the identification of C1q, a protein of the complement system, as the most bound biomarker to the surface of Gd@C_82_(OH)_22_ nanoparticles. Binding of C1q Gd@C_82_(OH)_22_ nanoconstructs led to the disruption of the secondary structure of the protein, nanoparticle endosomal internalization, and enhanced activation of the immune system. This approach could be further explored for the design of nanoformulations in which the nature of the personalized protein corona could be exploited for the production of novel diagnostic and anticancer nanomedicines.

Another aspect that can prevent nanoparticles cellular uptake is represented by the surface charge. The phospholipidic nature of cell membranes determines a general surface negative charge that can inhibit the uptake of anionic nanoparticles. On the other hand, positively charged nanoparticles could benefit from a better intracellular uptake, even if some cytotoxic effects have been reported [102]. The most common mechanisms of nanoparticles cellular uptake are represented by direct diffusion or endocytosis. The former entry mechanism is typical of nanostructures with a diameter of <5 nm [103], for lipid nanoparticles [104], and for nanoparticles grafted with cell-penetrating peptides, which are short aminoacidic sequences that help a covalently or non-covalently-bound cargo to penetrate inside a cell or a specific organelle [105]. This entrance route is usually desired when the nanoparticle payload must be released directly into the cytoplasm, such as for siRNA delivery [106].

During the last years, increased knowledge of endocytic processes allowed the design of “smarter” nanoparticles with improved targeting properties [13,107]. Nanoparticles with a size ranging from 100 to 500 nm are usually internalized through a clathrin-dependent endocytosis pathway [108,109]. This mechanism is triggered by the binding of nanoparticle ligands to specific cellular receptors. Ligand–receptor binding determines the formation of vesicles coated at the cytoplasmic level by clarithrin molecules whose polymerization brings vesicles to maturation and subsequent scissure from the cell membrane. The newly formed vesicles are later transported in the cytoplasm and uncoated from clarithrin, bringing about the formation of endosomes [110]. Similarly, a second common nanoparticle internalization pathway is represented by the caveolin-dependent endocytosis. This process does not require a ligand–receptor binding mechanism for its activation and the resulting endosomal vesicles are transferred to cytoplasmatic organelles, such as the endoplasmic reticulum and the Golgi apparatus. Caveolin-dependent endocytosis is usually observed for nanoparticles with sizes ranging from 50 to 100 nm [108,109].

Nanoparticles can also enter cells by phagocytosis and macropinocytosis. The first mechanism is carried out by phagocytes, immune cells that recognize elements foreign to the organism. The phagocytic process involves the recognition of such elements after binding to scavenger receptors. Phagocyte entrapment of the foreign materials brings about the formation of phagosome vesicles that are later fused to lysosomes, bringing about the formation of phagolysosomes [109]. Macropinocytosis is an actin-dependent non-specific mechanism of cellular uptake regulated by the activity of Ras protein [111]. Through this process, macrophages and dendritic cells can internalize viruses, growth factors, and particles whose size falls in the micromolar range. Macropinocytosis represents an important mechanism exploited by cancer cells for the translocation and trafficking of components of the plasma membrane and growth factors, contributing to the enhancement of cancer aggressiveness and metastasization [112]. Moreover, mutations in proteins of the Ras family, such as KRAS, are associated with a heightened macropinocytotic activity that comes up with cell proliferation and sustained ATP accumulation and consumption in the tumor microenvironment [113].

After cellular uptake, nanoparticles entrapped in endosomes through the abovementioned processes need to be released from these vesicles in order to exert their activity at the desired site of action. Rational modification of nanoparticles surface charge and material composition could be performed in order to disrupt the endosomal membrane and facilitate the release of the cargo [114]. In addition to this, endosomes are also characterized by an acidic environment. In this context, several examples of pH-sensitive nanoparticles have been reported [115,116,117,118], and the nature of the chemical bond that links the payload to the nanoparticle can be exploited for the pH-triggered release of the drug [119]. Finally, enzyme-cleavable bonds can also be utilized for the release of the therapeutic drug. Indeed, endosomes and lysosomes possess enzymes that can cleave specific chemical bonds or linkers properly inserted in the engineered nanoparticle in order to facilitate the release of the drug to the cytoplasm. For instance, the insertion of a Gly–Phe–Leu–Gly peptidyl linker has been exploited for the development of poly (glycolic acid) (PGA)–paclitaxel nanoparticles [120,121]. After the cleavage of the peptidyl linker by the lysosomal cathepsin-B enzyme, the drug can be easily released from the nanoparticle, exerting a better cytotoxic effect in NSCLC patients. Moreover, the estrogen-mediated higher production of cathepsin-B [122] led to better therapeutic results in women treated with such PGA-paclitaxel nanoparticles, paving the way for a gender-based personalized anticancer treatment.

In the context of precision medicine, nanoparticles also need to exert their effects at the subcellular level [123]. Nuclear drug delivery is often hampered by the size of the nuclear pore complex, which allows the passive diffusion of nanoparticles with a diameter of <10 nm [124]. For nanoparticles with a higher size, an active nuclear transport is required. This problem can be overcome by the insertion in the nanoconstructs of basic rich aminoacidic sequences known as nuclear localization signal (NLS) motifs [125], such as the trans activator of transcription (TAT) peptide (YGRKKRRQRRR), which bind to importins and translocate inside the cell nucleus [126]. This approach is particularly desired when the nanoparticle cargo is represented by a small molecule targeting DNA, such as doxorubicin.

Mitochondria represent a difficult subcellular organelle targeted by nanoparticles, mostly because of its highly negative membrane electric potential. In order to bypass this drawback, mitochondrial localization signal (MLS) can be used for the design of more efficient nanomedicines [127]. MLS are short peptide sequences containing basic and positively charged amino acids that can facilitate the entry of the cargo linked to the nanoparticle inside the cell. In addition, triphenyl phosphonium is often used for engineering mitochondria-targeting nanoparticles because of its lipophilic properties and its positive charge, with both properties favoring the internalization of the payload into the mitochondria [128].

Finally, the Golgi apparatus represents another potential subcellular nanoparticle target exploitable for the treatment of several pathologies, including cancer. This subcellular complex is responsible for the post-translational modifications and trafficking of proteins to different intracellular compartments. The alteration of Golgi’s activity could lead to the inactivation of proteins responsible for the onset of cancer. Intrigued by the observation that chondroitin sulphate accumulates in the Golgi apparatus of tumor cells and retinoic acid alters the Golgi apparatus morphology, Li and co-workers designed a nano formulation based on paclitaxel, chondroitin sulphate, and retinoic acid (PTX-CS–RA), which showed anti-metastatic effects by inhibiting metastasis-associated proteins through Golgi apparatus disruption and reduced tumor growth in 4T1 cells [129]. Similarly, Luo et al. reported the preparation of chondroitin-modified lipid nanoparticles loaded with doxorubicin (DOX) and retinoic acid (RA), which displayed better antitumor effects in SMMC–7721 hepatoma cells when compared to the separate administration of the two free drugs [130].

### 3.6. Overcome MDR Mechanisms

Multidrug resistance (MDR) phenomena represent a major drawback for the achievement of optimal therapeutic effectiveness. These mechanisms, usually related to the failure of cancer treatments, take into consideration a wide plethora of expedients utilized by cancer cells to inhibit cell death and sustain tumor progression, such as reduced drug cellular uptake, mutation of cellular targets, and increased drug inactivation through enhanced metabolism or alteration of drug targets [131,132]. The increasing knowledge in pharmacoproteomics, pharmacogenomics, pharmacogenetics, and pharmacometabolomics could be helpful in defining novel useful approaches for personalized therapeutic regimens [133]. In this context, nanomedicine and personalized medicine are strictly intertwined. Indeed, nanoparticles represent powerful tools in which it is possible to combine at the same time more than one drug, diagnostic agents, and/or biotechnological drugs with the aim to interfere with those mechanisms involved in the onset of MDR, even in a personalized manner.

One of the most important mechanisms related to MDR is represented by the overexpression of molecules transporters, better known as ATP-binding cassettes. These transporters extrude drugs outside the cell, lowering the optimal concentration of the drug required for the cytotoxic effect. Furthermore, cells who acquire resistance to chemotherapeutic agents usually become resistant also to drugs belonging to different chemical classes or to drugs acting with a different mechanism of action, globally worsening the MDR phenomena [28]. Pharmacogenomics studies focused on MDR-associated proteins are important in forecasting the goodness of a therapeutic strategy in a subset of patients in which interindividual differences could play a significant role in the success of a pharmacological approach [134,135].

The most known transporter involved in MDR is P-gp. This protein is responsible for the removal of doxorubicin, paclitaxel, etoposide, and vinblastine from cancer cells [136]. Nanomedicine approaches targeting the P-gp are usually based on the contemporary administration with a singular nano formulation of a cytotoxic drug and a P-gp inhibitor, such as verapamil, cyclosporine, or curcumin [136]. An additional strategy is also represented by the administration of siRNA-targeting genes involved in the production of ABC transporters [132,137,138]. The global result would consist of a reduced efflux of the drug outside the cell and its consequent higher intracellular concentration. For example, Jiang and co-workers reported the fabrication of RGD peptide-modified cationic liposomes delivering doxorubicin and ABCB1 siRNA [139]. Binding of the RGD peptide to integrin receptors of tumor cells enhanced the intracellular uptake of doxorubicin and siRNA. In vivo studies performed in a mouse model of doxorubicin-resistant MCF/A cells showed that these liposomes possessed a higher cytotoxic effect when compared to liposomes loading doxorubicin alone. This result should be attributable to the accumulation of the siRNA inside the cells with consequent higher cytotoxicity due to the administration of doxorubicin.

Genetic variants can also determine poor chemotherapy responses. Mutations in proteins involved in apoptotic processes, over-expression of pro-apoptotic, and down-regulation of anti-apoptotic proteins represent additional mechanisms involved in the onset of MDR. For instance, the over-expression of anti-apoptotic proteins belonging to the Bcl-2 family is one of the most common hallmarks of resistant cancer cells. Yu et al. reported on the co-delivery of epirubicin and Bcl-2 siRNA (siBCL-2) through pH-sensitive lipid nanoparticles [140]. The acidic endosomal environment allowed the escape of the siBCL-2 and proper tumor cell transfection; moreover, the lipid nano construct was also able to down-regulate P-gp overexpression and inhibited cell proliferation. Similarly, Ghaffari and co-workers designed a polyamidoamine (PAMAM) dendrimer loaded with curcumin and grafted with a Bcl-2 siRNA with improved anticancer effects in HeLa cells when compared with the effects exerted by curcumin alone or PAMAM-curcumin [141]. On the basis of these examples, it is obvious that an in-depth analysis of genetic variants and gene mutations through pharmacogenomics and pharmacoproteomic studies is highly desirable. Indeed, precise tumor typing could allow a better comprehension of the mechanisms involved in MDR, and consequently, it could be helpful in the design of personalized therapeutic strategies.

The tumor microenvironment (TME) and cancer stem cells also play a pivotal role in the development of MDR. Cancer stem cells are characterized by quiescence, they express drug efflux proteins, and possess an intrinsic resistance towards cytotoxic drugs because they frequently repair any possible damage in the DNA structure [142]. In addition, the tumor microenvironment contributes to cancer cell proliferation through the production of growth factors [143]. Due to their high plasticity after radiotherapy and chemotherapy, cancer stem cells can adopt a different phenotype that allows them to survive and give rise to a new subpopulation of tumor cells [144,145]. The identification of specific TME and cancer stem cells biomarkers is of particular importance for new personalized anticancer therapies [146,147]. Within this framework, Gaio and co-workers designed hyaluronic acid-coated polymeric nanoparticles for the delivery of docetaxel and the photosensitizer meso-tetraphenyl chlorine di-sulfonate targeting breast cancer stem cells over-expressing the CD44 glycoprotein, combining within a single nano construct chemotherapy and photodynamic therapy [148]. Binding of hyaluronic acid to CD44 allowed the nanoparticles to enter inside the cells through an endocytotic pathway, followed by cytotoxicity, which was enhanced by the presence of the photosensitizer. In another work, the hypoxic environment in which cancer stem cells reside was exploited for the design of hypoxia-sensitive nanoparticles [149]. A nitro-imidazole-modified hyaluronic acid–oxalate–camptothecin polymer-drug conjugate loaded with a differentiation-inducing agent, all-trans-retinoic acid (ATRA), was engineered to suppress MCF-7/CD44+ tumor growth. In tumor cells, reactive oxygen species (ROS) production triggered the disassembly of the nanoparticle with a consequent release of both ATRA and camptothecin. In cancer stem cells, the binding between CD44 and hyaluronic acid caused nanoparticle cellular uptake, whereas the hypoxic environment determined ATRA release but not camptothecin disassembly from the hyaluronic acid polymer. Binding of ATRA to retinoic acid receptors and cell differentiation also occurred. This last event determined a higher mitochondrial activity and ROS generation, which in turn brought about the release of camptothecin. In conclusion, the hypoxia-dependent release strategy allowed a controlled release in both cancer and non-cancer stem cells, with a reduction of drug resistance phenomena in the former and an overall tumor growth suppression and potential metastasization for both groups of cancer cells.

### 3.7. Solubility

The discovery of a new drug and its potential usage for the treatment of a certain pathology is often hindered by the physicochemical properties of the drug itself. In fact, highly lipophilic drugs with high molecular weight are often characterized by poor solubility that could prevent their utilization with consequent drug formulation issues [150]. Drugs endowed with high lipophilicity are not perfectly absorbed because they could be trapped in the phospholipidic bilayer of cells. On the other hand, hydrophilic drugs such as proteins and nucleic acids cannot be uptaken by simple passive diffusion by cells because of their incapacity to cross the cell membrane and could suffer poor stability in the aqueous environment [151]. Highly charged drugs such as DNA, miRNA, and siRNA could be administered after encapsulation in polymeric nanoparticles made of cationic building blocks [151]. These problems could be overcome by the encapsulation of such drugs into nanoparticles with a consequent improvement of their bioavailability.

5-Fluorouracil (5-FU) is a hydrophilic compound belonging to the chemical class of cytotoxic antimetabolites. Drug resistance phenomena have been encountered with this compound in colon cancer, and several strategies, such as the design of mutual prodrugs [152,153] or encapsulation in nanoparticles, are used to avoid this complication. For instance, 5-FU hydrophobicity and cytotoxicity can be alleviated by the production of prodrugs that can be efficiently loaded in xylan-stearic acid conjugates [154], liposomes [155] or exosomes [156]. These strategies alter the hydrophilic/hydrophobic nature of the molecule, allowing also a co-administration with compounds possessing different chemical profiles, such as doxorubicin or miRNAs.

The solubility of lipophilic compounds can be increased by loading the drugs in amphiphilic or hydrophobic nanoparticles. Thanks to this approach, Karve and co-workers revived the use of wortmannin, a phosphoinositide 3-kinase inhibitor whose clinical translation was hampered due to its high toxicity, poor stability, and high lipophilicity [157]. Loading of wortmannin in a biodegradable lipid–polymer nanoparticle platform reduced the intrinsic toxicity of the drug and enhanced its radio-sensitizing properties in vitro and in vivo.

Romana et al. suggested the utilization of liposome–micelle hybrids for the delivery of poorly soluble compounds. Using lovastatin as a drug model, they obtained better drug loading when compared with loading in traditional liposomes or micelles; in addition, they demonstrated a higher intestinal drug absorption and better transportation in a Caco-2 cell monolayer model through P-gp transporter inhibition [158].

Another drug with great potential in anticancer therapies is represented by curcumin. This natural compound is highly lipophilic and photo sensible, unstable in acidic and basic conditions, and rapidly metabolized and eliminated from the organism. The development of curcumin nanoparticles has been extensively reported [159,160,161,162,163] and the best results in terms of curcumin drug loading, stability, and solubility were reported by Gupta and co-workers through encapsulation in solid–lipid nanoconstructs [164].

Salinomycin, an anticancer antibiotic, is another drug possessing a low aqueous solubility. Ni et al. engineered PEGylated poly (lactic-co-glycolic acid) nanoparticles loaded with salinomycin and conjugated with a CD133 aptamer [165]. The new nanoformulations displayed a selective targeting and toxicity to osteosarcoma cancer stem cells expressing the CD133 protein, with the potential to overcome MDR phenomena linked to the pro-tumorigenic activity of such cancer stem cells.

Overall, these examples demonstrate that poorly soluble drugs can be reproposed after encapsulation in specific engineered nanoconstructs for the establishment of novel potential nanomedicine approaches. Moreover, grafting these nanoparticles with ligands targeting a specific receptor over-expressed in a subpopulation of cells could represent a suitable strategy to be further explored for personalized medicine therapeutic regimens.

## 4. Nanoparticles in Pharmacogenetic Testing

Pharmacogenetic testing is the first step for personalized medicine. In this respect, nanomaterials such as metal nanoparticles, dendrimers, liposomes, quantum dots, and carbon nanotubes have been explored for the assembly of a patient-specific molecular profile for providing an accurate diagnosis of specific targets/genes. These nanoparticles allow for testing a plethora of patient-specific genes, clinically leading to a high precision diagnosis and personalized management. Single DNA molecules can be sequenced using nanodevices and nano-systems due to the small dimension of nanoparticles. Further to a specific diagnosis, nanoparticles can provide a valuable tool to target the specific genetic abnormalities. Currently, it is possible to clinically test multigene panels for the patient and determine the decision on treatment for a particular drug [166].

Metallic nanoparticles have been studied in the diagnostic area of personalized medicine. Various nanoparticles such as gold nanoparticles (AuNPs), silver nanoparticles (AgNPs), iron nanoparticles (FeNPs), and other polymeric nanoparticles were investigated in pharmacogenetics.

The following is an overview of the usage of nanoparticles in personalized medicine on both diagnostic and therapeutic fronts (Table 1 and Table 2).

### 4.1. Gold Nanoparticles

Gold nanoparticles (AuNPs) are currently available as an integral part of biomarkers assays in the detection of various genetic abnormalities. AuNPs with dense DNA shells provide a versatile and programable means for the diagnostic purpose that can detect a wide range of genes. The gold nanoparticle part in the nano-bio-complex provides the optical advantages in the bioassay system with the ability of detecting the limit at the picomolar concentration [209].

In addition, AuNPs can be used as a fluorescence quencher to suppress the effect of luminescence or to enhance the electrochemiluminescence (ECL) of cadmium sulphide (CdS) nanocrystal. Chen et al. devised a hybrid system that utilizes AuNP with CdS nanoparticles for high-sensitivity detection of SNP. In this system, AuNPs are used as fluorescence quenchers to suppress the effect of luminescence or to enhance ECL of the CdS nanocrystal fabricated as a film through the distance modulation between the nano-metallic and semiconductor components by a hairpin DNA [167].

Surface plasmon resonance (SPR) has been utilized for the detection of SNP. For example, Jiang et al. [168] devised a system to detect TP53 point mutations utilizing oligonucleotides immobilized on AuNP. The system was inexpensive and could be readily used for clinical diagnosis. The same approach has been adopted by several companies for the detection of SNPs in multiple genes such as BRCA1 and cystic fibrosis (CF) genes [169,170].

Another advantage of AuNPs is their ability to measure intracellular gene expression. AuNPs can be constructed into “off-on” probes to quench light efficiently, resulting in lower background signals than conventional molecular approaches. An important property of the AuNPs DNA complex is their ability to be taken up by various types of cells.

Lee and co-workers [171] used this approach to detect the expression of heparin in cancer cells. Fluorophores were combined on the AuNP surface, resulting in a quenched fluorescence in an “off-state”. Upon a selective interaction with the expressed gene in metastatic cancer cells, the fluorophore is released, leading to a measurable fluorescent response. These same properties can be utilized to target and modify a specific gene intracellularly as a personalized therapeutic approach, as was explored by Seferos et al. [210].

Das et al. used the electrochemical properties of AuNP for the detection of specific tumor circulating DNA in the blood stream obtained from the patient’s serum, using the electrochemical chip-based method. Cancer cells with Kirsten rat sarcoma (KRAS) and BRAF mutations could be readily quantified with high sensitivity and specificity. This system is used as well to identify BRAF mutations in melanoma patients and to compare the outcome results of this chip, employing the electrochemical assay with a PCR-based method. The AuNP-based method achieved high levels of sensitivity and specificity with a 30 min short detection period advantage over the 2–3 h needed for a PCR [173]. Such a system can be of great value in identifying patients who would respond to Sotorasib, which specifically targets KRAS mutation and is used for the management of lung cancer [172].

In addition, the genotyping of large numbers of SNPs in an automated and a highly productive manner was achieved by Song Li et al., who developed a micro array system based on AuNP. The DNA primer-coated AuNPs were fabricated as nanobeads with a fluorophore and set on a clean glass slide; then each sample genotype was discriminated by bead array scanning. The system was applied on 320 samples for the detection of the C677T polymorphism of methylenetetrahydrofolate reductase (MTHFR) gene in a simple, fast, and cost-effective diagnostic tool [211]. This assay would be of great importance to predict the response to methotrexate in patients with rheumatoid arthritis and hematological malignancies and to predict the response to 5-fluorouracil in patients with colorectal cancer [174].

### 4.2. Silver Nanoparticles (AgNPs)

Silver nanoparticles are another attractive tool for pharmacogenetics testing and SNP detection. Various techniques employed for SNP studies can be augmented by AgNPs for ultrasensitive detection. AgNPs can infer high conductivity and enhanced electron transfer properties. Wu et al. devised a AgNP/Pt hybrid to fabricate a nanocluster probe with locked nucleic acid. The system was able to detect variant gene alleles in β-Thalassemia [175].

Jio et al. reported the electrochemical-sensing properties of AgNP combined with carbon nanotubes in the detection of SNP. In their study, they detected the SNP related to mitochondrial DNA mutation in type 2 diabetes mellitus by the DNA-mediated growth of AgNPs within the single-walled carbon nanotubes SWCNTs-modified electrode. This system accomplished high sensitivity to the targeted DNA with a low detection limit of 3 pM [176].

In another work Shi et al.used AgNPs to enhance the sensitivity and markedly improve the detection limit of surface-enhanced Raman spectroscopy (SERS), utilizing the dissolved silver ions from AgNPs. They forecast their system to be utilized in the future for ultrasensitive genetic studies [212]. In another study, Wabuyele et al. applied the plasmonic effect and SERS in the design of AgNPs label. This system exhibited high specificity and selectivity of AgNPs probes in the detection of single variation presence in the breast cancer BRCA1 gene [177]. The mutation in BRCA gene in ovarian cancer patients highly affects the choice of treatment and could determine their response to anticancer drugs such as platinum-based chemotherapeutics, and anthracyclines [213,214,215].

Another approach of the nanoparticle-based chemiluminescent (CL) method applied the AgNPs for ultrasensitive detection of SNP. The assay in this system depends on the hybridization of AgNPs with DNA. The hybridized technique was based on DNA-AgNPs probes coating polystyrene microwells. This was followed by the detection of the presence of the specific sequence DNA targets through the signal appearance after HNO3 solution addition. The system offers the advantages of quantification of target DNA, simplicity, and high sensitivity [178].

### 4.3. Quantum Dots (QDs)

QDs are NPs that have a range size between 1 and 20 nm and form metal salts (such as cadmium sulfide (CdS), cadmium telluride (CdTe), cadmium selenide (CdSe), and zinc oxide (ZnQDs). QDs pose multiple innate characteristics that are suitable for pharmacogenetic testing. QDs have a broad absorption spectrum, tunable emission, high quantum yields, and long lifetime fluorescence. These properties make them an ideal candidate for the design of nanoparticle probes adopting Förster resonance energy transfer (FRET). FRET is a phenomenon that happens when there is an energy transfer between two excited donors through non-radiative dipole–dipole coupling when they are in the proximity of a few nanometers [216,217].

QDs have been integrated into multiplexed SNP genotyping systems and used as fluorescent probes, such as the Qbead™ system developed by Xu et al. Qbead system was designed for the identification of around 200 SNP genotypes of cytochrome P450 family from clinical samples. The Qbead system was a reliable and highly sensitive system with 100% accurate results concordant with those obtained from direct DNA sequencing [179].

This system is particularly important to detect variants of CYP2D6 and CYP2C19 that can determine drug response variability in variants associated with ultrarapid metabolism compared to other variants associated with poor drug metabolism [218].

Another interesting genotyping QD-based system reported by Karlin-Neumann et al. employs four different QDs. The QDs were used for labeling in a microarray detection system, more than 10,000 SNPs from the unamplified DNA in a single reaction. The system proved advantageous in terms of broadband spectrum absorption with shorter wavelengths that resulted in higher levels of fluorescence emission [180].

QDs have been explored as fluorescent probes for the detection of gene mutations in chronic hepatitis B. Cheng Zhang et al. used QDs-mediated fluorescent method for the detection of hepatitis B M204I mutation that is associated with drug resistance. This mutation is important in HBV drug management with tenofovir dipivoxil, telbivudine, adefovir, lamivudine, and entecavir analogues [181].

For the detection of HBV mutants, the team utilized QD-labeled DNA. Then a fluorescence microscope was used to visualize the fluorescence signal emitted from the QDs with an excitation wavelength of 460–550 nm. The limit of detection for the detection of HBV genetic variation in this system was 103 IU/mL [219].

### 4.4. Iron Oxide Nanoparticles (FeNPs)

Iron oxide nanoparticles are members of the class of ferrimagnetic materials that can be used in different biomedical applications [220]. SNP detection using FeNPs has been explored in several studies due to the biocompatibility, high efficiency, simple and fast detection, reusability, and lack of interaction with separated substances activity [221].

Nam et al. reported the design of a sensitive scanometric assay of DNA-based NPs tagged with biological barcoded DNA that serves as signal probe, and oligo-DNA FeNPs serving as a capture probe for the targeted DNA to identify SNP variants. This system represents an ultrasensitive SNP sensing technology with a detection limit of attomole concentration (10^−18^ M) [182,183].

Liu et al. developed a high-throughput system for SNP genotyping utilizing solid-phase polymerase chain reaction using a biotin-labeled primer captured and amplified on the surface of streptavidin-coated FeNPs. Then SNPs were investigated by hybridization with probes. The system was tested to detect C677T polymorphisms of the MTHFR gene, yielding highly specific and sensitive results with the possible application of SNP on various targeted genes [184].

In addition, FeNPs probes can be employed for the simultaneous fluorescence and magnetic resonance imaging of targeted overexpressed genes. The FeNPs could be covalently or non-covalently bound to fluorescent dyes as reported by Wang et al. They used FeNPs coated with poly-amidoamine dendrimers that were conjugated to fluorescein isothiocyanate (FITC) and folic acid, applying a layer-by-layer assembly method for specific targeting of overexpressed folic acid receptors (FAR) in cancer cells. The FeNPs core was employed in conjunction with the FITC label to demonstrate their selectivity in targeting KB cells through magnetic resonance and fluorescence imaging [185].

### 4.5. Polymeric Nanoparticles (Polymeric NPs)

Polymeric NPs can be sorted into two types: nano capsules as reservoir systems, and nanospheres as matrix systems. They can be synthesized by natural materials, such as protein-based (albumin, gelatin, and collagen) polymers or polysaccharide-based (hyaluronic acid and chitosan) polymers. The other group of polymers is synthetic polymers, for example, dendrimers, PLGA, polyglycolic acid (PGA), polyacrylic acid (PAA), and polylactic acid (PLA).

Conjugated polymers (CPs) are large, repeated units, delocalized molecular structures that manifest some unique properties such as optical (colorimetric or fluorometric) and electrochemical properties, making them suitable as a sensing element. Several studies utilized CPs as transducers for SNP genotyping by FRET between CPs and a chromophore-labeled DNA probe. Other studies have utilized CPs for SNP detection through the conformational changes of CPs induced by a combination of specific DNA targets [222].

Duan et al. reported SNP genotyping assays using multi-step FRET and optical amplification of CPs. Water-soluble cationic polyelectrolytes can form complexes with DNA by electrostatic interactions [186]. The system design depends on the energy transfer cascade between complexes of CPs with two separately labeled DNAs with fluorescein and Cy3. The amplification of fluorescence signals is based on the CPs–DNA strongly electrostatic interaction. This system was successful in concomitant identification of three types of SNP genotypes in one extension reaction with high sensitivity.

In another study employing the optically amplifying CPs with the targeted DNA reported by Gaylord et al. [187], SNP detection and genotyping assays were applied in the detection of SNPs in chromosome 17 polymorphism associated with frontotemporal neurodegenerative disorders. Poly[(9,9-bis(6′-*N*,*N*,*N*-trimethylammoniumhexylbromide)fluorene)-co-phenylene] in addition to targeted DNA were recognized by sequence-specific hybridization between the fluorescein-labeled peptide nucleic acid probe and the target DNA sequence.

In a study by Li et al., the semiconducting fluorescent polymer (polyhedral oligomeric silsesquioxanes (POSS) was coated with poly(lactic-co-glycolic acid) (PLGA). The nanoparticles’ sizes ranged in between 230 and 260 nm, and their surface was functionalized with the antibody of human epidermal growth factor receptor 2 (HER2). This system was successfully used to discriminate between Her2-expressing SK-BR-3 cancer cells from cancer cells not expressing the abovementioned receptor through fluorescence with high quantum yield [188].

## 5. Challenges in Nanomedicine and Personalized Medicine

There has been a rapid increase in the number of nanomedicines that are being developed as therapeutics, however, clinical translation efficiency is less than satisfactory, which indirectly suggests and highlights different challenges related to nanomedicine that need to be addressed before introducing them for translational use. The major issues with the nanomedicines are related to their safety at a biological level, their cost, and upscale wide range production. Similarly, precision medicine, which relates to identifying and targeting an individual’s genome for a particular disease, also faces challenges related to ethical, social, and legal issues [223]. Nanomedicines, when administered to the body, encounter the biological environment, but there are no standard safety evaluation parameters that determine their toxicity. Moreover, morphological and physiochemical characteristics can affect the biodistribution and interaction with the biological membranes when in systemic circulation, and hence, pose a threat to safety. There may be issues associated with specific nanomedicine products and for which particular evaluations might be required. Toxicity in the medical industry is defined in terms of acute and chronic toxicity. While the acute toxicity refers to inflammation, hemolysis, oxidative stress, and other organ-specific effects as such, the chronic toxicity occurs after a long period of time and is not usually addressed [224].

There is a need to understand the physiochemical properties of both the drug and the nanocarrier, along with the reagents and pathways utilized for synthesis, from a toxicological perspective for improving their chances of success. As opposed to traditional pharmaceuticals, nanomedicine is a complex three-dimensional system with multiple components, each of which is intended to serve a specific purpose. Due to their complexities, advanced analytical tests are needed to fully detect, characterize, and quantify each component, as well as evaluate the interactions between them and the interaction between the drug-loaded nanoparticle and the biological systems. Furthermore, an additional issue of concern is the stability and the long-term storage of these nanoparticles-based products; hence, the type of lipids and polymers used for synthesis must be kept in check [225]. On a larger scale, even after a nanomedicine is introduced into market, it can be withdrawn if the quality and safety levels do not comply with the standard regulations.

To translate nanomedicine into a personalized drug entity, large-scale synthesis with high reproducibility is necessary. Typically, these drug-loaded nanocarriers are synthesized in small batches in laboratories, or for preclinical and clinical investigations. The large-scale production on the other hand is much more difficult considering the complexities of different nanoparticles, and even the slightest change in the manufacturing process can cause critical alterations in the characteristic’s properties of these nanoparticles. It is crucial that parameters such as the ratio of the nanocarrier polymer, entrapped drug, organic solvent, crosslinker, emulsifier, and even temperature, and pH be determined and kept at appropriate levels. In addition to this, the process of scaling up of these nanomedicines is often a costly multistep process not just in terms of manufacturing but also in terms of the cost needed for preclinical and clinical studies, which makes a new nanomedicine unlikely to succeed. It can be challenging to gain regulatory approval for new nanomedicines, especially if existing products with similar efficacy already exist with the same indication, hence the researchers’ work to improve the biodistribution, bioavailability and efficacy of conventional drugs [226]

Moreover, developing nanomedicines in general is hampered by a lack of standards and regulations in manufacturing practices, quality control, safety, and efficacy evaluation. Now, the standards of nanomedicines are determined by regulatory authorities such as USFDA and EMA to keep a check and to provide guidance, although no specific regulatory standards governing their production or clinical translation have been devised till date. Therefore, when it comes to the application of nanotechnology, different geographical regions have distinctive approaches such as the nanomedicine approved in one country might not get approval in the other [227,228].

Similarly, when participating in drug testing in personalized medicine, patients will need to understand the potential risk that might be associated with it, and hence receiving informed consent can be a rigorous process in various cases. Moreover, just like the production of nanomedicine, the cost related to the development of a precision medicine is also very high and will require funding for years with a daunting uncertainty of success. The technologies required for sequencing of an individual’s DNA and designing nanomedicine also add to the cost. Therefore, prior to introducing personalized medicine in the market, it must become part of a routine healthcare. There is a need to understand the disease at the molecular and genetic level, to interpret the genetic testing required for the patient, and then design the medication thereafter, which will not only reduce the chances of failure but will also improve the knowledge and understanding of a patient’s specific conditions [229,230,231].

Regarding clinical trials, it is important that these trails be performed as individualized treatment plans to cope up with the ever-changing landscape of diseases.

In order to enroll patients in clinical trials, multicenter collaboration is critical when drug development is in its early stages. Additionally, there are established concepts in the field of drug development, such as phase 1 expansion cohorts replacing phase 2 tests, regulatory approvals based on nonrandomized trials, and tumor-agnostic approvals. Yet, in clinical trials, there are problems related to patient accrual and unresolved regulatory issues, as well as technical limitations of molecular tests that need to be resolved [232,233].

Although these advanced treatments are still relatively new, they have already shown promise in treating conditions such as epilepsy, cystic fibrosis, and some forms of cancer and diseases that have a deeper genetic basis. There are two different ways by which precision medicine can change clinical trials. Firstly, more trials need to be conducted to test the efficacy of medicine in patients, which is almost like what happens in MATCH trails. Secondly, gene sequencing must be performed simultaneously to help in creating and dividing patients into different subtypes of disease.

It has been suggested that smaller clinical trials can serve to be a better and more efficient option when testing a precision medicine. Scientists have proposed that when a drug is meant for a small group of targeted individuals, it would show variation in terms of the efficacy and the chances of it failing in a regular clinical trial would be high. For instance, the drug ivacaftor (Kalydeco) meant for cystic fibrosis treatment was approved in 2012 for certain specific gene mutations and was effective in only about 4% of the total patients of cystic fibrosis. Clinical trials done in smaller specific populations might as well produce efficacious results, and hence, faster drug approval. Researchers are still working on defining and designing better randomized studies to improve the translation of precision medicine in the market [234]. In order to integrate genomic data efficiently and dynamically and to evaluate the validity of matching unique genomic alterations with specific interventions or treatments, new strategies have been developed. These include clinical studies with adaptive designs, umbrella trials, basket trils, and aplatform trials among others. Adaptive design refers to a clinical study designed to adapt to data analysis (usually intermediate findings) of the study subjects with the intention of modifying one or more specified elements of the research design and hypothesis. Similarly, an umbrella trial, also known as a master trial, is a program in which patients are identified as eligible based on the presence of a specific tumor type that is sub stratified based on specific molecular alterations matched with different anticancer therapies. For example, this method was utilized in the MoTriColor EU H2020-funded project, which involved a set of molecularly guided trials with specific treatment strategies in patients with advanced newly molecular defined subtypes of colorectal cancer. Unlike the above-stated methods, in a basket trial, different tumor types with a common molecular alteration are treated with the same matched therapy. Through these approaches, targeted agents can be evaluated in molecularly selected populations and are accessible for patients across a wide range of tumor types, potentially including rare cancers that could not be studied in conventional trials. Lastly, the platform trials serve as an effective method, since this trial aims to evaluate different therapeutics and/or target cohorts for one or more diseases through ongoing changes in sub studies [235,236,237].

## 6. Conclusions

The new field of research in personalized/precision medicine is emerging, utilizing a precision, personalized diagnosis of a genetic profile of an individual and targeted treatment of that disease condition. It considers the genetic, phenotypic, and environmental factor of a patient or a group of similar patients that could have an influence on the safety and efficacy of a particular treatment. Over recent years, researchers were active in combining different aspects of nanotechnology for their use in personalized medicine.

Nanomedicine has opened new avenues not just for the delivery of drugs, but equally in molecular and genetic diagnosis.

Personalized medicine can exploit nanomedicine for increasing binding affinity; achieving better bioavailability and compatibility; and having maximum therapeutic efficacy with a controlled drug release profile for the drug to reach the right target, in the right patient, at the right time. It can also contribute to understanding an individual genome and thereby designing endpoint strategies for diagnosis and therapeutics.

Laboratory-synthesized nanoparticles have shown promising results and it could just be a matter of time before nanomedicine is used at a far greater scale in personalized medicine. Both nano and personalized medicine have a wide scope and can collectively be the future of medicine.

## Figures and Tables

**Figure 1 jpm-12-00673-f001:**
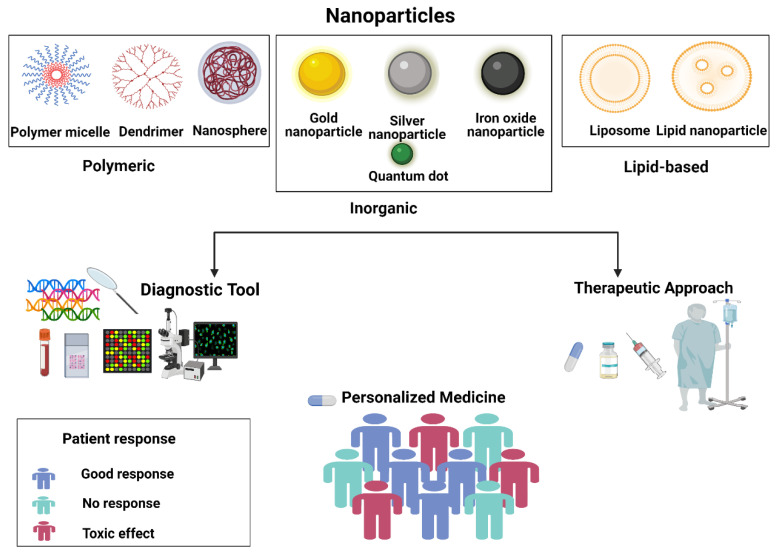
A schematic representation of nanotechnology used in personalized medicine.

**Figure 2 jpm-12-00673-f002:**
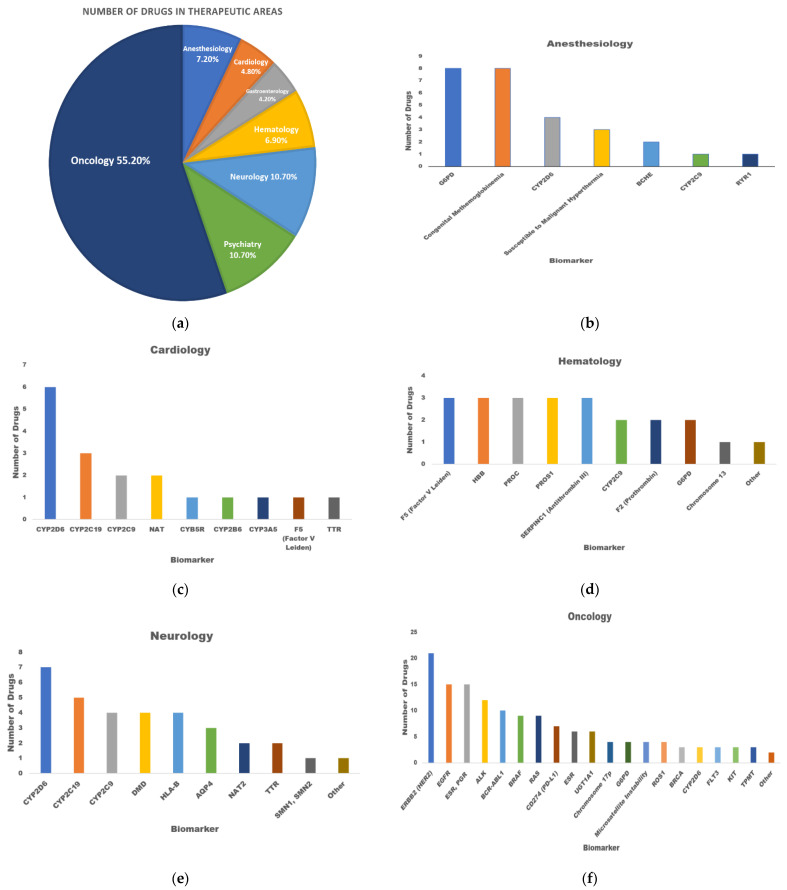
(**a**): Number of drugs in therapeutic areas; (**b**): biomarkers related to clinical therapeutics area anesthesiology; (**c**): biomarkers related to clinical therapeutics area cardiology; (**d**): biomarkers related to clinical therapeutics area hematology; (**e**): biomarkers related to clinical therapeutics area neurology; and (**f**): biomarkers related to clinical therapeutics area oncology.

**Figure 3 jpm-12-00673-f003:**
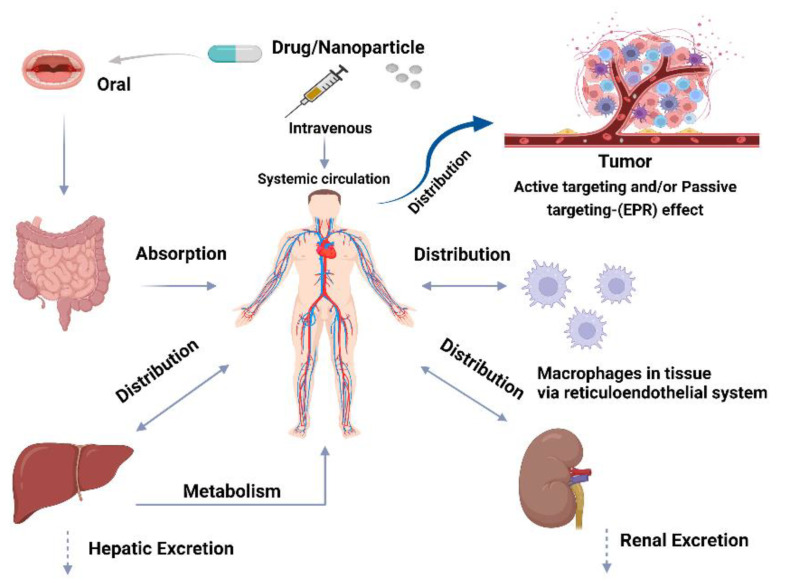
Pharmacokinetics absorption, distribution, metabolism, and excretion (ADME) of personalized drugs or drugs combined with nanoparticles.

**Figure 4 jpm-12-00673-f004:**
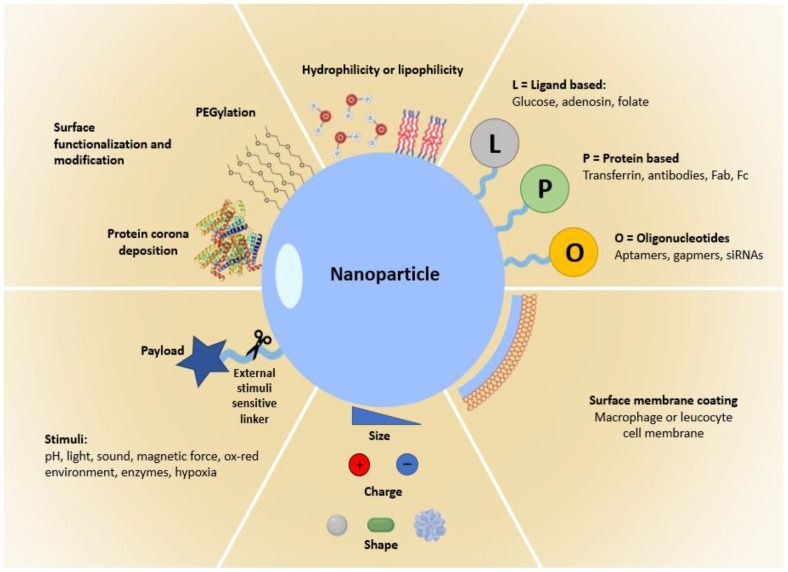
Overview of nanoparticles modifications to improve PK and PD properties.

**Table 1 jpm-12-00673-t001:** Summary of nanotechnology application in genetic testing.

Nanoparticles	Diagnostic	Targets	References
AuNPs			
	AuNPs are used as fluorescence quenchers	detection of SNP	[167]
	AuNP	detect TP53 point mutations	[168]
	AuNP	detection of SNPs in BRCA1	[169]
	AuNP	detection of SNPs in CF genes	[170]
	AuNPs probes	detect the expression of heparin in cancer cells.	[171]
	AuNPs electrochemical chip-based method	Detection of cancer cells with KRAS and BRAF mutations in lung cancer	[172,173]
	AuNPs fabricated as nanobeads with fluorophore in micro array system	for the detection of C677T polymorphism of MTHFR gene	[174]
AgNPs			
	AgNP/Pt hybrid fabricated as nanocluster probe	detect variant gene alleles in B-Thalassemia	[175]
	AgNP combined with carbon nanotubes	detect the SNP related to mitochondrial DNA mutation	[176]
	AgNPs probes	detection of single variation presence in the breast cancer BRCA1 gene	[177]
	DNA-AgNPs probes coating polystyrene microwells	detection of the presence of the specific sequence DNA targets	[178]
QDs			
	QDs Qbead system	multiplexed SNP genotyping systems of 200 SNP genotypes of CYPP450 family	[179]
	QDs labelling in a microarray detection system	10,000 SNPs from the unamplified DNA in a single reaction	[180]
	QDs-mediated fluorescent method	detection of hepatitis B M204I mutation, which is associated with drug resistance.	[181]
FeNPs			
	FeNPs scanometric assay of DNA-based NPs	identify SNP variants	[182,183]
	Biotin label captured and amplified on the surface of streptavidin-coated FeNPs	to detect C677T polymorphisms of MTHFR gene	[184]
	FeNPs coated with poly-amidoamine dendrimers conjugated to fluorescein isothiocyanate and folic acid	targeting of overexpressed FAR cancer cells	[185]
Polymer NPs			
	cationic polyelectrolytes form a complex with DNA by electrostatic interactions	identification of three types of SNP genotypes in one extension reaction	[186]
	optically amplifying Poly[(9,9-bis(6′-*N*,*N*,*N*-trimethylammoniumhexylbromide)fluorene)-co-phenylene] with the targeted DNA	SNP detection and genotyping assays were applied in detection of SNPs in chromosome 17 polymorphism associated with frontotemporal neurodegenerative disorders.	[187]
	fluorescent polymer (polyhedral oligomeric silsesquioxanes) with PLGA with the surface antibody of HER2.	distinguish the high Her2-expressing cancer cells	[188]

Single nucleotide polymorphism (SNP), cystic fibrosis (CF), Kirsten rat sarcoma (KRAS), B-Raf proto-oncogene (BRAF), methylenetetrahydrofolate reductase (MTHFR), CYP P450 cytochrome P450 family, folic acid receptor (FAR), human epidermal growth factor receptor 2 (HER2), and poly (lactic-co-glycolic acid) (PLGA).

**Table 2 jpm-12-00673-t002:** Summary of drugs/nano construct applications in therapeutic areas and their targets.

Nanoparticles	Therapeutic	Targets	Reference
AuNPs			
	Afatinib conjugated to AuNPs	EGFR in NSCLC	[189]
	self-assembly gefitinib conjugated to colloidal AuNPs	EGFR to treat lung cancers	[190]
	Dasatinib loaded on AuNPs	CML	[191]
	(PEG-PPG-PEG) with functionalized AuNPs tyrosine kinase inhibitor- Vandetanib, (ZD6474).	EGFR and VEGFR—for treatment of metastatic breast cancer	[192]
AgNPs			
	AgNPs embedded in graphene oxide conjugated with the folate analog, MTX	folate receptor-positive breast cells	[193]
	Capecitabine bonded to AgNPs	Antiproliferative and proapoptotic effects for different cancers	[194]
	AgNPs/FeNPs modified with (PEG)-carboxyl and folate and loaded with DOX	cancer cells	[195]
QDs			
	erlotinib conjugated to QDs	EGFR in NSCLC	[196]
	carbon quantum dot CQD-based DOX nanocarrier system	against breast cancer cells	[197]
	CQD system conjugated with Quinic Acid loaded with gemcitabine	targeting agent toward breast cancer	[198]
	graphene quantum dots with imatinib	decrease BCR-ABL activity by targeting ABL, c-kit, and PDGF-R-treatment of leukemia	[199]
FeNPs			
	erlotinib-conjugated FeNPs	EGFR in NSCLC	[200]
	Erlotinib-conjugated FeNPs	lung adenocarcinoma	[201]
	FeNPs–carbon nanotubes with (PAMAM–PEG–PAMAM) linear-dendritic copolymers loaded with DOX	hybrid nanostructure can be used for targeting, imaging, and cancer treatment	[202]
	dasatinib-loaded FeNPs core with self-assembly micelles	multitargeted inhibitor of many essential kinases impacting oncogenesis in breast cancer	[203]
Polymer NPs			
	poly (α, l-glutamic acid) polymer/selumetinib and dabrafenib	BRAF, MEK—melanoma	[204]
	SMA/Crizotinib and dasatinib	Met, ROS1, KIT, and ABL—glioblastoma multiforme	[27]
	SMA/Sorafenib and nilotinib	VEGFR, PDGFR, FLT3, ALK, FGFR, c-KIT, JAK, CSF1R, RET, and Bcr-Abl—prostate cancer	[205]
	chitosan-based polymeric nanoparticles/Imatinib	Bcr-Abl—colorectal cancer	[206]
	PLGA polymer/Tamoxifen	estrogen receptor-positive breast cancer cells	[207]
	PLGA polymer/Erlotinib	EGFR in NSCLC	[208]

Chronic myeloid leukemia (CML), epidermal growth factor receptor EGFR, non–small-cell lung cancer (NSCLC), amphiphilic polymer of polyethylene glycol-block-polypropylene glycol-block-polyethylene glycol-block (PEG-PPG-PEG), vascular endothelial growth factor receptor (VEGFR), methotrexate (MTX), amino-poly (ethylene glycol) (PEG), doxorubicin (DOX), stem cell factor receptor (c-kit), platelet-derived growth factor receptor (PDGF-R), polyamidoamine–polyethylene glycol–polyamidoamine (PAMAM–PEG–PAMAM), B-Raf proto-oncogene (BRAF), Mitogen-activated protein kinase (MEK), poly(styrene-co-maleic acid) (SMA), anaplastic lymphoma kinase (Met), Proto-Oncogene 1, Receptor Tyrosine Kinase (ROS1), Tyrosine-protein kinase (ABL), Fms-like tyrosine kinase-3 receptor (FLT3), anaplastic lymphoma kinase (ALK), fibroblast growth factor receptor (FGFR), Janus kinase (JAK), colony-stimulating factor 1 receptor (CSF1R), and glial cell line-derived neurotrophic factor receptor(RET).

## Data Availability

Not applicable.

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
