# Peer review of "The Promise of Nanotechnology in Personalized Medicine"

_jpm, 2022, doi:10.3390/jpm12050673_

Round 1
Reviewer 1 Report
Article Name: The Promise of Nanotechnology in Personalized Medicine
Manuscript ID: jpm-1669790
Recommendation: Major Revision as noted.
Comments:
General Comments: The review focus on The Promise of Nanotechnology in Personalized Medicine. The writing of review and presentation of the article is poor. In summary, I have a number of general concerns, followed by a range of specific comments, which prevent me from recommending this paper.
Comments:
- There are a number of reviews/articles on nanomaterials for medicine related applications. In this context, it is important that the authors here clarify what is new content and what is overlapping between this manuscript and other recent reviews. Here are some examples.
https://link.springer.com/article/10.1007/s13346-020-00819-z
https://www.ncbi.nlm.nih.gov/pmc/articles/PMC2813556/
https://www.frontiersin.org/articles/10.3389/fchem.2018.00360/full
- The manuscript discuss more recent article.
- The author should rework the manuscript to provide a prospective on the literature reviewed.
- The quality of figures is not good.
Author Response
We would like to thank the editor and the reviewers for their time, effort, and thoughtful comments on our manuscript.
Please find our detailed responses to reviewers' comments.
- There are a number of reviews/articles on nanomaterials for medicine related applications. In this context, it is important that the authors here clarify what is new content and what is overlapping between this manuscript and other recent reviews. Here are some examples.
We thank the reviewer for the comment. We agree that many review articles had tackled the same topic. In our plan for the manuscript, we made sure that we don’t overlap with earlier reviews. The current review is unique regarding:
- Focusing on the current state of personalized medicine in clinical application.
- Discussing in detail the Pharmacokinetics and pharmacodynamics advantages of the nanomedicine for personalized medicine
- Discussing the role of nanoparticles in pharmacogenetics.
E.g.: https://link.springer.com/article/10.1007/s13346-020-00819-z
The review focus was on the microneedle arrays-mediated technology application in cancer.
https://www.ncbi.nlm.nih.gov/pmc/articles/PMC2813556/. The article is focusing on nanotechnology applied to medicine and dentistry.
https://www.frontiersin.org/articles/10.3389/fchem.2018.00360/full
This article is focused on the introduction of nanomedicine in the pharmaceutical market and the challenges for nanotechnology implementation and the issues pertaining to the applications.
Thus overall, we think the current structure is unique.
- The manuscript discuss more recent article.
We thank the reviewer for the comment, but we strived to discuss as much recent relevant references to the review. Almost 50% of the cited articles are published within the last 5 years.
- The author should rework the manuscript to provide a prospective on the literature reviewed.
We added the scope of our review in the introduction and highlighted it in yellow (line 60-63)
- The quality of figures is not good.
We improved the quality of the figures as well adding a new figure (figure 4)
Reviewer 2 Report
Comments to the authors:
Figure 1. Please enlarge the font, it is difficult to read the captions on the picture.
Row 69: Please replace personalized drug with personalized medicine.
Figure 2. Please enlarge the font, it is difficult to read the captions on the picture.
Figure 3. Please replace semicolon: Active targeting:pasive targeting with and/or
Line 252 and line 253: Please use uniform full names for the abbreviations: fragment antigen-binding (Fab)-252, antigen-binding fragment (Fab)-253.
Lines 366-372: How protecting the drug from the degradation enzymes in in the patient having a poor metabolizer phenotype, as reported for the CYP2C9*2 and CYP2C9*3 variant allele will influence/protect patient from concentration-related neurotoxicity? Sounds contradictory. Please give other example or explanation.
Line 433: Replace clarithrin with clathrin
Line 533: Delete one dot.
Line 611: Replace comma with is.
Author Response
We would like to thank the editor and the reviewers for their time, effort, and thoughtful comments on our manuscript.
Please find our detailed responses to reviewers' comments.
- Figure 1. Please enlarge the font, it is difficult to read the captions on the picture.
We thank the reviewer for the comment, Font has been modified.
- Row 69: Please replace personalized drug with personalized medicine.
Personalized drug has been replaced with personalized medicine.
- Figure 2. Please enlarge the font, it is difficult to read the captions on the picture.
Font has been modified.
- Figure 3. Please replace semicolon: Active targeting:pasive targeting with and/or
We thank the reviewer for the observation, corrected as per the reviewer’s recommendation.
- Line 252 and line 253: Please use uniform full names for the abbreviations: fragment antigen-binding (Fab)-252, antigen-binding fragment (Fab)-253.
Abbreviations have been modified.
- Lines 366-372: How protecting the drug from the degradation enzymes in in the patient having a poor metabolizer phenotype, as reported for the CYP2C9*2 and CYP2C9*3 variant allele will influence/protect patient from concentration-related neurotoxicity? Sounds contradictory. Please give other example or explanation.
We thank the reviewer for the comment and apologize for the oversight. We changed the example into patients with fast metabolizing phenotype to fit the added value of nanomedicine. The change is line 472-479.
- Line 433: Replace clarithrin with clathrin
We thank the reviewer for the comment, clarithrin has been replaced with clathrin.
- Line 533: Delete one dot.
We thank the reviewer for the comment, dot deleted
9 Line 611: Replace comma with is
Corrected
Reviewer 3 Report
The manuscript by Alghamdi et al. reviews topics from the personalized medicine and nanomedicine area, and proposes the personal nanomedicine approach. The concept of personalized nanomedicine is novel, but is still in its infancy. It’s understandable that not much research has been conducted yet, and this review seems to discuss each section mostly independently. Despite this, it’s encouraged to find the intersection of personal medicine and nanomedicine as much as you can. It’s important to build the relationship between your sections by connecting them. Some comments are listed below.
1. The section 3 was discussed generically and similar summaries have been repeatedly reported in review articles. Much of this knowledge is known. It’s important that this section include the information of how individual physiological differences affect the PK/PD of nanomedicine, and how the PK/PD of nanomedicine can be personalized. This section is also quite long and is recommended to be shortened.
2. It’s worthwhile to look for drugs summarized or discussed in section 2 and see which of them have been used in nanomedicine applications. Such discussions could be very valuable to the topic.
3. This review needs to stand out from existing literature on similar topics, focus on what are new developments and be different and more updated than previous reviews. Just to list a few below.
https://www.ncbi.nlm.nih.gov/pmc/articles/PMC5748624/
https://www.transbiomedicine.com/translational-biomedicine/personalized-nanomedicine-not-just-a-tool-but-towards-an-excellence.php?aid=9820
https://aacrjournals.org/clincancerres/article/18/18/4889/77364/Personalized-NanomedicinePersonalized-Nanomedicine
https://www.sciencedirect.com/science/article/pii/S1359644617302362
https://www.futuremedicine.com/doi/10.2217/nnm.15.152
4. Check spelling errors in figures and texts.
Author Response
We would like to thank the editor and the reviewers for their time, effort, and thoughtful comments on our manuscript.
Please see our detailed responses to reviewers' comments.
- The section 3 was discussed generically, and similar summaries have been repeatedly reported in review articles. Much of this knowledge is known. It’s important that this section include the information of how individual physiological differences affect the PK/PD of nanomedicine, and how the PK/PD of nanomedicine can be personalized. This section is also quite long and is recommended to be shortened.
We thank the reviewer for the comment. We added accordingly example of the advantage of nanomedicine under absorption (line 159-164); distribution (line 204-209). Other examples have been highlighted in yellow. While we acknowledge the reviewer comment to shorten the section, we think it can be more valuable with the current details.
- It’s worthwhile to look for drugs summarized or discussed in section 2 and see which of them have been used in nanomedicine applications. Such discussions could be very valuable to the topic.
We agree that the discussion of the drugs mentioned in the table could improve the value of the table. Indeed, we had this part written (7 extra pages), however we realized that the review would be enormous in size (currently 44 pages), so we opt to only use the table to keep the focus of the review.
- This review needs to stand out from existing literature on similar topics, focus on what are new developments and be different and more updated than previous reviews. Just to list a few below.
We thank the reviewer for the comment. We agree that many review articles had tackled the same topic. In our plan for the manuscript, we made sure that we don’t overlap with earlier reviews. The current review is unique regarding:
- Focusing on the current state of personalized medicine in clinical application.
- Discussing in detail the Pharmacokinetics and pharmacodynamics advantages of the nanomedicine for personalized medicine
- Discussing the role of nanoparticles in pharmacogenetics.
E,g, , https://www.ncbi.nlm.nih.gov/pmc/articles/PMC5748624/
This article focuses on current nanomedicines in pre-clinical and clinical development, challenges to turn nanomedicine into personalized medicine, and also the process of development of novel nanomedicines from their design in research labs to the market.
https://www.transbiomedicine.com/translational-biomedicine/personalized-nanomedicine-not-just-a-tool-but-towards-an-excellence.php?aid=9820
This article is a rapid communication, with an overall view and no specific detailed discussion.
https://aacrjournals.org/clincancerres/article/18/18/4889/77364/Personalized-NanomedicinePersonalized-Nanomedicine
This article focusses on the specific application of nanomedicine as a theragnostic tool to provide optimum anticancer management.
https://www.sciencedirect.com/science/article/pii/S1359644617302362
This article is focusing on the Central nervous system (CNS) disease therapeutics challenges and solutions by using the nanotechnology.
https://www.futuremedicine.com/doi/10.2217/nnm.15.152
This article is an editorial with generalized discussion rather than focused review.
Thus overall, we think the current structure is unique.
4. Check spelling errors in figures and texts.
Done.
Reviewer 4 Report
This review focus on the recent advances of nanotechnology and their applicability in the field of personalized medicine. Undoubtedly, nanotechnology is a very promising tool to treat many diseases and recent advances in material science proposed nanoparticles and nanostructures that can be functionalized with therapeutic agents, including drugs and nucleic acids, which could be combined with other treatments currently employed to offer improved solutions for the clinical practice.
The review is overall well-structured and covers several aspects of the employability of nanomaterials for targeted and precision medicine. However, some part should be improved as following:
- The authors should report some examples of clinical trails using nanotechnology for personalized medicine. Also, it should be mentioned the limiting aspects of such trails and how they could be potentially improved (for example: targeting others pathways, chemical modifications of nanostructures.. ect)
- It is absolutely true that nanoparticles can improve the pharmacokinetics and pharmacodynamic proprieties of drugs, and this is likely the most important advantage using them; it would be interesting to stress more on the possible modification of nanoparticles to allow a better interaction with the cell and eventually the microenvironment. Some recent literature should be cited DOI: 10.1038/s41573-020-0090-8 and a scheme or figure may help to summarize the potential chemical modification.
- The impact of protein corona in nanoparticles targeting and therapy is mentioned several times. This important reference should be included and briefly discussed for completeness: DOI 10.1038/s41467-017-00600-w
- The major weakness of the review is the lack of discussion regarding the therapeutic use of nanoparticles, with annexes examples. I would expect that after “Nanoparticles for genetic testing”, the discussion follow with “Nanoparticles for disease treatment ”… or similar . The examples are reported in Table 2, those should be described, at least in part, within the text. This will improve the readability of the manuscript for the readers.
- This referee considers important resalt application in cancer therapy, since the relevance of tumor diseases. In particular, modified nanoparticles can delivery many therapeutic agents, like siRNAs, gapmers, or aptamers to target key genes involved in cancer progression and therapy resistance. Some literature for some functional examples could help this discussion: DOI: 10.3390/pharmaceutics13122067 ; 10.1016/j.ymthe.2021.05.004 ; 10.3390/cancers13164038
Author Response
We would like to thank the editor and the reviewers for their time, effort, and thoughtful comments on our manuscript.
Please see our detailed responses to reviewers' comments.
1- The authors should report some examples of clinical trails using nanotechnology for personalized medicine. Also, it should be mentioned the limiting aspects of such trails and how they could be potentially improved (for example: targeting other pathways, chemical modifications of nanostructures. ect) .
We thank the reviewer for the comment. We have added a new part under section 5, line 1140 to 1183. The section is highlighted in yellow. As for the clinical trials listing, this was mentioned in other reviews such as in:
https://www.ncbi.nlm.nih.gov/pmc/articles/PMC5748624/
we think it will be repetitive to add this to the current review.
2- It is absolutely true that nanoparticles can improve the pharmacokinetics and pharmacodynamic proprieties of drugs, and this is likely the most important advantage using them; it would be interesting to stress more on the possible modification of nanoparticles to allow a better interaction with the cell and eventually the microenvironment. Some recent literature should be cited DOI: 10.1038/s41573-020-0090-8 and a scheme or figure may help to summarize the potential chemical modification.
We thank the reviewer for the comment. We discussed the new section (3.4) as per the reviewer’s comment. Line (495 -582). We also included the reference suggested by the reviewer.
3- The impact of protein corona in nanoparticles targeting and therapy is mentioned several times. This important reference should be included and briefly discussed for completeness: DOI 10.1038/s41467-017-00600-w
In response to the reviewer's comment, we added a discussion highlighting the above point, Line 602- 623). We also included the reference suggested by the reviewer.
3- The major weakness of the review is the lack of discussion regarding the therapeutic use of nanoparticles, with annexes examples. I would expect that after “Nanoparticles for genetic testing”, the discussion follow with “Nanoparticles for disease treatment ”… or similar . The examples are reported in Table 2, those should be described, at least in part, within the text. This will improve the readability of the manuscript for the readers.
We agree that the discussion of the drugs mentioned in the table could improve the value of the table. Indeed, we had this part written (7 extra pages), however we realized that the review would be enormous in size (currently 44 pages), so we opt to only use the table to keep the focus of the review. Nonetheless, we mentioned examples of nanoparticles to explain the advantages of Nanomedicine in section 3. All the examples are highlighted in yellow.
4- This referee considers important resalt application in cancer therapy, since the relevance of tumor diseases. In particular, modified nanoparticles can delivery many therapeutic agents, like siRNAs, gapmers, or aptamers to target key genes involved in cancer progression and therapy resistance. Some literature for some functional examples could help this discussion: DOI: 10.3390/pharmaceutics13122067 ; 10.1016/j.ymthe.2021.05.004 ; 10.3390/cancers13164038
In response to the reviewer's comment, we added a discussion regarding the therapeutic agent’s delivery, like siRNAs, gapmers, or aptamers highlighting the above points, line (296 to 374) .
Round 2
Reviewer 1 Report
Accept
Reviewer 3 Report
The revised manuscript may be accepted for publication now.
Reviewer 4 Report
The authors improved the manuscript accordingly to the concerns raised by the reviewer.